# VibeCheck: Discover & Quantify Qualitative Differences in Large Language Models

**Lisa Dunlap**
UC Berkeley

**Krishna Mandal**
UC Berkeley

**Trevor Darrell**
UC Berkeley

**Jacob Steinhardt**
UC Berkeley

**Joseph Gonzalez**
UC Berkeley

## Abstract

Large language models (LLMs) often exhibit subtle yet distinctive characteristics in their outputs that users intuitively recognize, but struggle to quantify. These "vibes" – such as tone, formatting, or writing style – influence user preferences, yet traditional evaluations focus primarily on the singular vibe of correctness. We introduce **VibeCheck**, a system for automatically comparing a pair of LLMs by discovering identifying traits of a model ("vibes") that are well-defined, differentiating, and user-aligned. VibeCheck iteratively discovers vibes from model outputs and then utilizes a panel of LLM judges to quantitatively measure the utility of each vibe. We validate that the vibes generated by VibeCheck align with those found in human discovery and run VibeCheck on pairwise preference data from real-world user conversations with Llama-3-70b vs GPT-4. VibeCheck reveals that Llama has a friendly, funny, and somewhat controversial vibe. These vibes predict model identity with 80% accuracy and human preference with 61% accuracy. Lastly, we run VibeCheck on a variety of models and tasks including summarization, math, and captioning to provide insight into differences in model behavior. VibeCheck discovers vibes like Command X prefers to add concrete intros and conclusions when summarizing in comparison to TNGL, Llama-405b often overexplains its thought process on math problems compared to GPT-4o, and GPT-4 prefers to focus on the mood and emotions of the scene when captioning compared to Gemini-1.5-Flash. Code can be found at https://github.com/lisadunlap/VibeCheck

## 1 Intro

> **vibe check :** A process by which a group obtains a subjective assessment of another person, place, or thing.
> – *Urban Dictionary*

How a large language model writes a story, explains a concept, or edits an essay can be evaluated along many different dimensions such as creativity, formatting, and writing style. However, most evaluations focus on one dimension: *"correctness"*. State-of-the-art in evaluation methods remain largely focused on measuring accuracy for question answering and analytical reasoning tasks (Hendrycks et al., 2021a; Wang et al., 2019b;a; Hendrycks et al., 2021c), and methods which aim to provide a more holistic view of LLMs (Zhang et al., 2024; Padlewski et al., 2024; Mehri & Eskenazi, 2020b) rely on predefined concepts like conciseness, clarity, and trustworthiness to measure a model's performance. These evaluation approaches fail to capture the open-ended nature of LLM applications and the critical dependence on subjective user preferences and context of the task. For instance, tone and creativity might be crucial in creative writing, whereas efficiency and readability are crucial in coding tasks. To best inform users of which model would be best for their needs, we require flexible evaluation methods that can both *discover* and *measure* the relevant axes to evaluate for a given task.

When interacting with a set of LLMs for an extended period, a user can often tell which model generated a particular response by looking at certain traits of the outputs. We define these identifying traits of models as *"vibes"*. For instance, users have found Llama-3 outputs tend to be more friendly compared to outputs from GPT-4 and Claude which tend to be more formal (see Figure 1); in other words, Llama-3 ranks high on the friendliness vibe, defined by the axis formal → friendly. Using these insights, we might select Llama for customer service tasks and Claude for coding tasks.

Understanding these vibes helps inform the development and deployment of models, but discovering and validating them for each model can be time-consuming and difficult. To address this, we outline how one can find and, more importantly, measure an LLM's vibe by formalizing three necessary and quantifiable traits of a useful vibe: *well-defined* (agreement among multiple users), *differentiating* (ability to distinguish between models), and *user-aligned* (predictive of user preferences).

We introduce **VibeCheck**, a system which qualitatively analyzes pairs of models by automatically finding well-defined, differentiating, and user-aligned vibes. Motivated by recent work in using LLM's in lieu of human judgment (Zheng et al., 2023; Zhang et al., 2024; Zhong et al., 2023; 2022; Dubois et al., 2023), VibeCheck models the qualitative analysis process by identifying the axes on which these model outputs differ to obtain a core set of vibes (e.g friendliness). Once these vibes are obtained, VibeCheck employs a panel of LLM judges (Verga et al., 2024) to determine where each model's output falls on this vibe (e.g. more formal or more friendly) in order to obtain numeric scores which are then used to measure a vibe on each of our 3 key criteria.

We run VibeCheck on several datasets to evaluate its effectiveness across different scenarios in Section 5. First, we validate that the vibes discovered by VibeCheck align well with human-annotated differences between ChatGPT and human responses using the Human ChatGPT Comparison Corpus (HC3). Next, we demonstrate that VibeCheck outperforms a predefined list of vibes in predicting user preferences on real-world comparison data from Chatbot Arena, achieving 80% accuracy at predicting model identity and 61% accuracy and predicting user preference. Inspecting the vibes of VibeCheck, we find that Llama-70b uses more typographic emphasis, more examples, and is funnier than GPT-4 and Claude-3-Opus. Conversely, we find that GPT-4 and Claude comment much more on ethics and limitations than Llama, which is more willing to give controversial responses.

Lastly, in Section 6 we apply VibeCheck to several applications: text summarization on CNN/Daily-Mail, math problem-solving on MATH, and image captioning on COCO. Using VibeCheck, we find insightful qualitative differences between models with similar accuracy on correctness metrics but differing user preferences. For instance, Command X prefers to add concrete intros and conclusions when summarizing in comparison to TNGL, Llama-405b often overexplains its thought process on math problems, and GPT-4 prefers to focus on the mood and emotions of the scene when captioning.

## 2 RELATED WORK

**Aspect-based evaluations.** The number of benchmarks in the NLP community has exploded in recent years, with a growing body of work on exploring a more holistic evaluation of language models. Several works (Pang et al., 2020; Banerjee & Lavie, 2005; Sellam et al., 2020) aim to improve on automatic metrics like BLEU (Papineni et al., 2002) and ROUGE (Lin, 2004) scores to better measure how well a models output aligns with the ground truth by incorporating more nuanced evaluation criteria like factual accuracy, fluency, and conciseness. Similarly, efforts have been made (Liang et al., 2023; bench authors, 2023; Kiela et al., 2021; Wang et al., 2019b;a) to standardize model evaluation by evaluating models on many of these metrics across various tasks.

Moving away from measuring model outputs on ground truth responses, work from Mehri & Eskenazi (2020b); Zhang et al. (2024); Li et al. (2019); Mehri & Eskenazi (2020a); Gehrmann et al. (2021) evaluate model outputs on criteria like helpfulness and clarity using LLM judges on more open ended tasks like dialogue, role-play, and summarization. While these efforts supply a great foundation for measuring correctness, they all define the axes on what makes something correct beforehand. In contrast, VibeCheck aims to automatically discover these axes (vibes) and verify their utility to the user by measuring the correlation between vibes and human preference.

**Pairwise comparison of LLMs.** HCI tools like Google's AutoSxS (Google Cloud, 2024) and LLMComparator (Kahng et al., 2024) explores the current state of human powered LLM qualitative evaluation through interviews with data analysts. These works find that practitioners often eyeball individual examples to interpret and look at qualitative differences between the outputs of two models, and develop an interactive web based application for users to inspect side-by-side LLM outputs with an LLM based rationale as to why one output is preferred over another. While these works are focused more on software tools rather than a pipeline which can be quantavely verified, these HCI findings inform VibeCheck's vibe discovery mechanism to align with the human-powered qualitative process. Moreover, many NLP works (Zheng et al., 2023; Verga et al., 2024; Li et al., 2023; Park

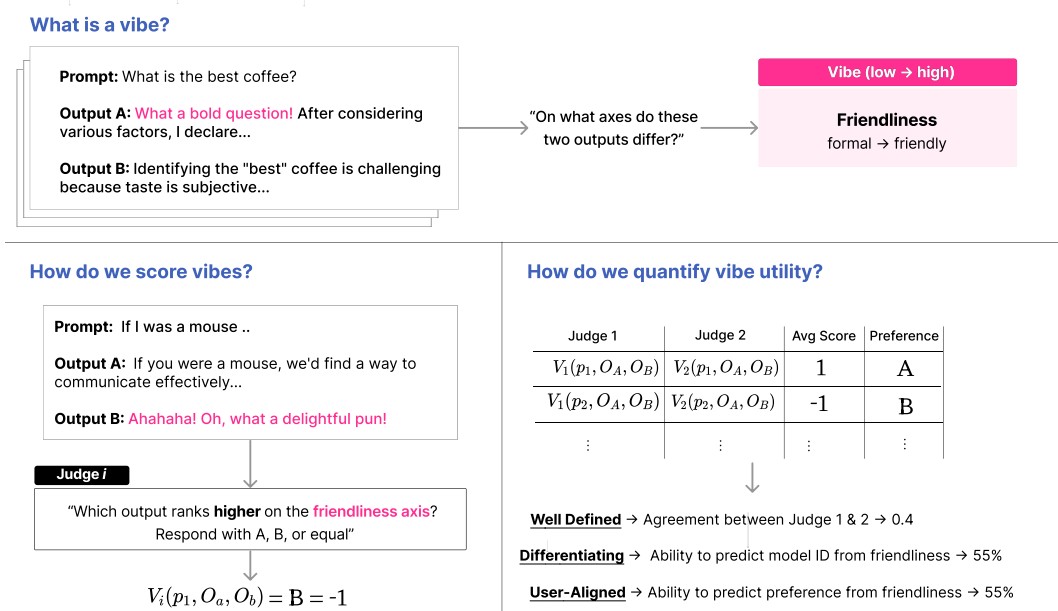

Figure 1: **Core components of VibeCheck.** A vibe is an axis along which a pair of outputs differ: for example, in the top panel, output A is more friendly while output B is more formal, defining a friendliness vibe. To score a prompt output triplet, a panel of LLM judges are used to determine which output falls higher on the vibe, resulting in a score of 1 (A), -1(B), or 0(tie). Finally, the scores obtained over a large set of outputs along with preference labels are used to compute vibe utility.

et al., 2024; Liusie et al., 2024) have explored using LLMs to predict user preference given responses from two models, showing these preference predictions often align with the judgements of human annotators. While these efforts focus more on the user experience, it does not provide an interpretable view of exactly *why* these users prefer one output over the other.

**Discovering separable traits in unstructured data.** In parallel to works in the machine learning community on LLM evaluation, there has been fantastic efforts in the HCI community on comparing generative model outputs as well as on using LLMs for qualitative analysis. Works like Torii et al. (2024); Byun et al. (2023) use LLMs to generate discussions from qualitative research data to automate the data analysis process, but note the lack of comprehensive evaluation metrics. Automated data analysis on unstructured data has also been explored in Zhong et al. (2022; 2023); Dunlap et al. (2024b), which use LLMs and VLMs to propose and validate candidate differences between two sets of text or images in the form of "set A contains more X", and Chiquier et al. (2024) employs an evolutionary algorithm to find text descriptions which best separates image classes to assist in zero-shot classification. We extend these works to pairwise inputs and introduce metrics of success which can better verify the separability, consistency, and alignment of these differences.

## 3   VIBE-BASED EVALUATIONS

We define a *vibe* as an axis along which a pair of texts can differ (e.g., "formal → friendly") that is perceptible to humans. A vibe $\nu$ is represented by a text description of the axis along with a definition of what it means to be high or low on this axis (e.g. "Tone: low = formal, high = friendly", see Figure 1). Identifying vibes aids users in selecting models that best suit their specific tasks. In this work, we focus on comparing the vibes of two models by discovering the axes on which their outputs differ and quantifying the utility of these vibes.

Consider a dataset $D$ composed of triples $(p, o_A^P, o_B^P)$ and preference labels $y_p$, where $p$ is a prompt and $o_i^P$ are the outputs from models $A$ and $B$. For each triple, a judge (human or LLM) assigns a score for vibe $\nu$, denoted $\nu(p, o_A^P, o_B^P) \in \{-1, 0, 1\}$, which indicates whether model $A$ scores lower (-1), similarly (0), or higher (1) than model $B$ on this vibe. Thus, a vibe imposes an ordering on model outputs.

We define 3 key criteria of a useful vibe; it should be *well-defined*, *differentiating*, and *user-aligned*.

**Well-defined:** multiple evaluators agree on the ordering of outputs along the vibe. We quantify this by having two different judges (typically LLMs) compute $\nu(\text{p}, \text{o}_A^\text{P}, \text{o}_B^\text{P})$ across dataset $D$ and report Cohen's Kappa to assess agreement.

**Differentiating:** one model's outputs consistently rank higher on this vibe compared to the other's across a set of prompts. We quantify this by calculating a *separability score* for each vibe, which measures how consistently the vibe distinguishes between the two models across all samples.

$$\texttt{sep\_score}(\nu) = \frac{1}{\mid D \mid} \sum_{\text{p} \in D} \nu(\text{p}, \text{o}_A^\text{p}, \text{o}_B^\text{p})$$

To measure separability across a set of vibes, we fix a pair of models $(A, B)$ and measure the accuracy of using $\nu(\text{o}_A, \text{o}_B)$ to classify which output came from which model. We also more generally measure separability for a set of vibes $\nu_1, \ldots, \nu_k$, by using $\nu_{1:k}(\text{p}, \text{o}_A, \text{o}_B)$ as a $k$-dimensional feature vector, then training a linear classifier to predict model A vs. model B, and reporting accuracy on a held-out set. We refer to this metric as *model-matching accuracy*.

**User-aligned.** One potential use of vibes is to better understand human preferences. While a vibe like *"frequent use of the letter 'e'"* may be differentiating, it is unlikely predictive of human preferences. We assume our tuples $(\text{p}, \text{o}_A^\text{P}, \text{o}_B^\text{P})$ are annotated with user preferences $y \in \{-1, +1\}$, indicating which model's output is preferred. We train a logistic regression classifier to predict $y$ using the same feature set $\nu_{1:k}$ as above, reporting held-out accuracy. We refer to this metric as *preference prediction accuracy*. We can measure the influence of a single vibe on preferences by examining the coefficients and p-values of the preference prediction model.

VibeCheck automatically finds high-scoring vibes across the three criteria through an iterative process: (1) discovering vibes, (2) computing their scores, (3) selecting those meeting all criteria, and (4) focusing on tuples $(\text{p}, \text{o}_A^\text{P}, \text{o}_B^\text{P})$ where existing vibes fail to differentiate the two models. We repeat this process to extract new, more distinguishing vibes, thus optimizing for the three key criteria while continuously refining the set of vibes.

## 4 VIBECHECK

VibeCheck consists of 3 stages: vibe discovery, vibe validation, and vibe iteration. Further details on the method implementation and prompts used are located in the Section D.

**Vibe discovery.** Similar to how a data scientist would inspect a subset of examples to discover qualitative differences in outputs, we discover vibes by having an LLM (GPT-4o (OpenAI, 2024)) examine the differences seen in a random subset of $d$ prompt triplets. We first split the $d$ prompt triplets into smaller batches of size $batch$ and prompt GPT-4o to find differences between model $A$ and model $B$ across the set $\{(\text{p}_1, \text{o}_A^1, \text{o}_B^1), ..., (\text{p}_{batch}, \text{o}_A^{batch}, \text{o}_B^{batch})\}$. To encourage the vibes to be well-defined and user-aligned, we prompt GPT-4o to generate differences that are human-interpretable and informative for understanding the overall behaviors of A and B. Below is a paraphrased system prompt used by the proposer.

```
You are a machine learning researcher analyzing outputs from two
LLMs on the same input, identify differences along specific, mutually
exclusive, and clearly defined axes that are easily interpretable by
humans.  For each axis, provide a concise description of what it means
for an output to be "Low" and "High" on this axis.
```

An example axis generated in this step might be 'Tone: Low: formal; High: friendly'. We repeat this proposal step for $\lfloor d/batch \rfloor$ sets of triplets, obtaining a final set of vibes $\{\nu_1, .., \nu_M\}$ by taking the union of the vibes generated in each batch. We found that GPT-4o generates 5-10 axes of variation (vibes) for each sample, so we summarize vibes across all samples in $D_{\text{discovery}}$ to find a set of $K$ vibes which appear most often in $\{\nu_1, .., \nu_M\}$.

**Vibe validation.** Given a vibe $\nu$ from the discovery phase, we first apply each vibe to a set of validation tuples, then use this validation set to score vibes and compute inter-annotator agreement, model-matching accuracy, and preference prediction accuracy and filter out vibes with low scores.

To apply vibes on the validation set, we assign a score to each pair of outputs $\nu_j(p, o_A^p, o_B^p) \in \{-1, 0, 1\}$, indicating whether model $A$ scores lower (-1), similarly (0), or higher (1) than model $B$ on the vibe. A score of 0 is assigned if the outputs are equal on this vibe or if the vibe is not applicable (e.g., the vibe is about coding style but neither output contains code); otherwise, we compute the score using a set of LLM judges (GPT-4o-mini (OpenAI, 2024) and Llama-3-70b (AI@Meta, 2024)). We average the score of the 2 judges and then round to -1, 0, or 1 (so 0.5 is rounded to 1 and -0.5 to -1). To avoid position bias (Zheng et al., 2023), we run each LLM judge twice on each sample, swapping the order of the outputs. If the judge's decision is dependent on the position of the output, we deem this pair of outputs as having a similar vibe and assign a score of 0 for that judge.

Next, we use these scores to quantify each vibe on our 3 criteria and filter out any which are not well-defined, differentiating, and user-aligned. We ensure each vibe is well-defined by computing the inter-annotator agreement (Cohen's Kappa) for each $\nu_j$ across $D_{\text{validation}}$ and remove any with Cohen's Kappa less than 0.2, which indicates a weak agreement among judges. To ensure each vibe is differentiating, we compute the separability score and discard any vibes with a score below 0.05. As we explicitly prompt the model to produce vibes which provide useful insights into the behavior of language models, we assume these vibes are already aligned with users. Using the remaining $k$ features, we run logistic regression using the scores $\nu_{1:k}(p, o_A, o_B)$ as features to obtain our model matching and preference prediction models.

**Vibe iteration.** The filtered vibes generated in the initial vibe discovery set may not capture all the differences that contribute to user preference, resulting in a low model matching and preference prediction accuracy. We address this by iteratively refining our vibes based on tuples $(p, o_A^p, o_B^p)$ that were misclassified by our prior differentiation stages. Specifically, we take the prompt output triplets that were misclassified by the model matching model and ask an LLM to find new axes on which these misclassified prompts vary, which are also not represented in the current set of vibes. We then perform the same summarization/reduction procedure as before, run vibe validation/filtering, and append the resulting new vibes to the existing set of vibes. We repeat this process for a fixed number of iterations $i$. In practice we find that after 3-5 iterations the discovery process does not find any additional vibes that significantly reduce the error rate of the model matching predictor.

## 5 RESULTS

We first validate VibeCheck by comparing its discovered vibes to those identified by human annotators in Section 5.1. Next, we evaluate VibeCheck on real-world user-LLM conversations with pairwise preference data, measuring the vibes' well-defined, differentiating, and user-aligned through inter-annotator agreement, model matching accuracy, and preference prediction accuracy on a heldout set. In Section 5.2 compare the discovered vibes' performance against an predefined list of common qualitative analysis criteria. Lastly, in Section 6, we demonstrate VibeCheck's broader applicability by analyzing model differences across summarization (Hermann et al., 2015), mathematical reasoning (Hendrycks et al., 2021c), and image captioning (Lin et al., 2014; Chen et al., 2023).

**Experimental setup.** Unless otherwise stated, we run VibeCheck for 3 iterations, use a proposer batch size of 5, and set $D_{discovery}$ to be 20 samples per iteration. Some datasets such as MATH, CNN/DailyMail, and COCO captions have no pre-computed preference labels; to simulate preferences, we apply LLM-as-a-judge and ensemble GPT-4o and Claude 3.5 Sonnet as a judge using a similar procedure to (Zheng et al., 2023), removing any samples declared a tie. Additional details on the experimental setup and hyperparameters are given in the Section A.

We compute average Cohen's Kappa, model matching accuracy, and preference prediction accuracy on the top 10 vibes generated by VibeCheck on a held-out set of prompt tuples with preference labels. To obtain the top 10 vibes, we apply least-angle regression on the full set of vibes returned by VibeCheck to predict model identity, then sort by the separability score. The full list of vibes discovered, LR coefficients and p-values from the model matching and preference prediction models, Cohen's kappa per vibe, and separability scores are in the Section G.

**List of predefined Vibes.** As a baseline, we prompt GPT-4o to generate a set of 10 vibes shown in Figure 3 and Table 6 which represent common axes on which LLM outputs differ.

## 5.1 MEASURING VIBECHECK'S ALIGNMENT WITH HUMAN DISCOVERY

In this section, we compare the findings from VibeCheck to findings obtained via human discovery to ensure that the vibes discovered and measured by LLM's align with humans. We utilize previous work (Guo et al., 2023), which collects responses written by humans and GPT-3.5 (Schulman et al., 2022) for the same list of questions and then recruits 200 annotators to look at 100-200 prompt output triples presenting the characteristics of both human responses and ChatGPT answers. This results in a set of 10 insights (vibes) which are listed in detail in Section B.

In Table 1 we show a summarization of the top 10 vibes found by VibeCheck along with the corresponding insight found by humans which align with each vibe meaning. We see that VibeCheck uncovers most of the same vibes as the human annotators, aside from (1) GPT fabricates facts and (2) GPT focuses on a literal interpretation of the question while humans address different aspects of the question and can infer hidden meaning. The inability to find these vibes is likely a weakness of our GPT proposer, as these vibes relate to the inherent weaknesses of GPT. The complete table of VibeCheck outputs is located in Figure 7.

| VibeCheck Vibes | Human Discovered Vibes |
|---|---|
| Humans include more references and citations | Humans include detailed citations of papers and books. |
| GPT is more formal/academic, Humans are more casual/ conversational | GPT answers are typically formal, humans' are more colloquial |
| GPT includes disclaimers about advice limitations | GPT refuses to answer questions outside its knowledge |
| GPT is cautious to give advice, emphasizes seeking professional help | GPT shows less bias and harmful information |
| GPT has cohesive, fluid responses with clear sentence structure | GPT writes in an organized manner with clear logic |
| GPT is strictly informative, humans include personal anecdotes | GPT gives objective answers, humans use subjective expressions |
| GPT has less emotional engagement, humans' acknowledge emotions | GPT expresses less emotion, humans convey their feelings |
| GPT has longer, more informative responses | GPT has longer more detailed responses. |
| GPT has more thorough & detailed responses | GPT has longer more detailed responses. |
| GPT has more comprehensive responses | GPT has longer more detailed responses. |
| - | GPT is strictly focused on the question, humans diverge and shift topics |
| - | GPT may fabricate facts |

Table 1: **Comparison of VibeCheck vibes to human labels.** Complete table in Figure 7. We see that the vibes discovered by VibeCheck closely align with vibes found through human analysis.

## 5.2 DESCRIBING USER PREFERENCE ON CHATBOT ARENA

On April 18th 2024, Meta released their open-weight large language model Llama 3. On benchmarks like MMLU, Llama-3-70b outperforms Claude-3-Sonnet and Gemini 1.5. It had even stronger results on Chatbot Arena (Chiang et al., 2024), a popular platform for community-driven LLMs where users submit a prompt, receive responses from 2 anonymous models, and vote on which output they prefer. On this leaderboard, Llama-3-70b is ranked similarly to the top proprietary models like GPT-4 and Claude3-Opus. This has led to speculation on whether there are qualitative properties of Llama that make it popular among users (Dunlap et al., 2024a).

In this section, we analyze the qualitative differences between Llama-3-70b and other top models using pairwise comparisons from Chatbot Arena. We run VibeCheck on a set of combined battles (pairwise comparisons) between Llama-3-70b VS GPT-4 and Llama-3-70b VS Claude3-Opus[1] under three settings: using the entire dataset, and using 2 subsets of the data: STEM prompts (including coding) and Writing prompts, which include creative writing, humanities questions, and general chatting. We obtain these subsets by using GPT-4o-mini to categorize the questions as a STEM Question, a Writing/Chatting prompt, or neither. The size of each subset can be found in Section A.

We compare the vibes found by VibeCheck to a list of predefined vibes (Table 6) of common differences between language models which a user may be interested in. Table 2 shows that VibeCheck achieves higher model matching accuracy than the predefined vibes all categories and more iterations improve model matching and preference prediction accuracy. Furthermore, Figure 2 shows that the vibes are more fine-grained. We summarize our other findings below:

---

[1]Data: https://huggingface.co/datasets/lmarena-ai/Llama-3-70b-battles

**Comparing MM and PP accuracy across topics.** Table 2 shows that MM and PP accuracy is lower for STEM questions compared to writing or overall prompts. We suspect this is because Llama's qualitative traits (friendliness, humor, safety, etc.) are less relevant for objective questions like coding and math, and user preferences here are influenced more by factual accuracy than stylistic traits. Conversely, VibeCheck best predicts preferences for writing-oriented prompts, as style is often more important for these open ended tasks.

To understand how user preferences for these vibes vary across task domains and contexts, we analyze separability scores and preference prediction coefficients for predefined vibes in Figure 3. For writing tasks, formality, humor, and expressive emotional content positively correlate with user preference, while these traits negatively correlate with STEM tasks, where logical rigor is the most influential on preference. Conversely, logical rigor has minimal impact on preferences for writing tasks. While our dataset does not directly compare individual judgments, treating STEM and writing task users as distinct groups provides preliminary evidence of task-specific preferences. Additionally, lower separability scores for STEM tasks indicate less stylistic divergence in model outputs for objective questions like coding and math, making model identity harder to predict, consistent with Table 2.

**Notable vibes for Llama-3 70B.** The top 10 vibes uncovered by VibeCheck (Figure 2) highlight Llama's use of formatting, willingness to engage with sensitive topics, less emphasis on ethics, and a conversational, humorous style. Finer-grained vibes include Llama's use of bold/italics to emphasize points and increased use of personal pronouns, with 'I,' 'we,' and 'you' appearing 3x more in Llama outputs than GPT/Claude conversations. The preference prediction coefficients in Figure 2 show Chatbot Arena users tend to prefer outputs which are less focused on ethics, employ markdown and typographic emphasis to highlight key points, and employ humor to engage the user, all of which are vibes which llama possesses. We believe that this correlation between vibes and user preference can explain some of the discrepancy seen in llamas high ranking on the leaderboard in comparison to models like GPT-4 which often outperform Llama.

| Vibe (low -> high) | Sep Score [-0.4,0.4] | PP Coef [-0.5,0.5] | Cohen |
|---|---|---|---|
| **Language and Tone.** Professional, straightforward tone. **->** Enthusiastic, friendly tone. | | | 0.51 |
| **Typographic Emphasis.** Minimal use of typographic emphasis, letting the text stand alone. **->** Uses typographic emphasis like bold or italics to highlight key points. | | | 0.64 |
| **Interactivity.** Provides information passively without engaging the user. **->** Encourages user interaction, such as posing questions or suggesting actions. | | | 0.44 |
| **Formatting Completeness.** Responses are minimally formatted, relying on plain text. **->** Responses include comprehensive formatting, such as Markdown or additional stylistic elements. | | | 0.57 |
| **Examples and Illustrations.** Minimal examples. **->** Provides multiple examples. | | | 0.61 |
| **Use of Humor.** Maintains a serious tone without humorous elements. **->** Employs humor frequently to engage the reader. | | | 0.62 |
| **Use of Personal Pronouns.** Rarely or never uses personal pronouns. **->** Frequently uses personal pronouns (I, we, you). | | | 0.32 |
| **Ethical Consideration.** Provides factual information without commenting on ethics. **->** Offers ethical considerations in its responses. | | | 0.53 |
| **Humility.** Projects confidence and completeness without discussing limitations. **->** Frequently acknowledges limitations in the response or areas of uncertainty. | | | 0.41 |
| **Formality Level.** Uses informal or conversational language. **->** Uses formal language and expressions. | | | 0.45 |

Figure 2: **Comparing Llama-3-70b VS GPT-4 & Claude-3-Opus on Chatbot Arena.** Negative separability scores indicate Llama-3-70B aligns with the low (red) description, while negative preference coefficients show alignment with low descriptions is preferred. We see that Llama is more humorous, utilizes more formatting, provides more examples, and comments much less on ethics than GPT and Claude: all attributes which correlate positively with human preference.

## 6 APPLICATIONS

We next apply VibeCheck to discover qualitative differences between models' behavior on three open-ended tasks: text summarization, math problem-solving, and image captioning. We use CNN/DailyMail (Hermann et al., 2015) for text summarization, MATH (Hendrycks et al., 2021b) with chain-of-thought prompting for problem-solving, and COCO for image captioning. For CNN

| Method | Overall | | | STEM | | | Writing | | |
|---|---|---|---|---|---|---|---|---|---|
| | M.M. | P.P. | C.K. | M.M. | P.P. | C.K. | M.M. | P.P. | C.K. |
| VibeCheck [1 iter] | 68.68 | 60.00 | 0.42 | 65.20 | 55.99 | 0.44 | 74.09 | 60.58 | 0.51 |
| VibeCheck [3 iter] | **80.34** | 59.34 | 0.46 | **68.71** | 57.31 | 0.45 | **77.19** | **62.04** | 0.49 |
| Predefined Vibes | 72.10 | **61.11** | 0.51 | 65.94 | **58.38** | 0.45 | 75.00 | 59.49 | 0.52 |

Table 2: **Comparing Llama-3 to GPT and Claude on Chatbot Arena.** We report Model Matching Accuracy (M.M.), Preference Prediction Accuracy (P.P.), and average Cohen's Kappa (C.K) for the full dataset (Overall) and STEM and Writing categories. VibeCheck achieves higher model matching accuracy than Predefined Vibes and similar preference prediction accuracy. VibeCheck obtains the largest improvements over predefined vibes in the writing category, suggesting that for open-ended prompts, model styles differ significantly, and style has a greater influence on preference.

Figure 3: **Comparing user preference and separability across STEM and writing tasks.** On predefined list of vibes referenced in Table 2. Negative preference coefficients indicate a preference for low-description vibes, while negative separability scores show Llama responses align more with the low description than Claude or GPT responses. For writing tasks, detailed explanations, humor, and expressive emotion positively correlate with human preference, while these traits negatively correlate with STEM tasks. Conversely, logical rigor has a stronger positive impact on preference for STEM tasks. These trends are reflected in separability scores, with less separability on STEM tasks for vibes like humor and emotional tone, and more separability for logical rigor.

and MATH we use cached model predictions downloaded from HELM (Liang et al., 2023) and intentionally choose models which are ranked similarly to each other, but when running LLM as a judge to get preference labels, one model is more heavily preferred. For captioning, we generate captions on a random subset of 1000 COCO images. The vibes for each application in Section G.

## 6.1 WHAT DO DIFFERENT MODELS FOCUS ON WHEN SUMMARIZING?

We compare the summary styles of TNLG v2 (Smith et al., 2022) (530B) to Cohere's Command X large Beta (Inc., 2023) on the CNN/DailyMail dataset. While these models achieve a similar mean win rate on the HELM leaderboard, we see when using LLM as a preference judge, Command X has a win-rate of 71.12%. Looking at the top 5 vibes located in Figure 14, we find that (1) Command X clearly states an introduction and conclusion while TNLG utilizes choppy sentences without an either (2) Command provides specific examples or anecdotes to illustrate points and (3) Command is able to capture multiple viewpoints and emotional aspects of a story while TNLG is more objective. We see these qualities are positively correlated with human preference, which may explain the disparity

between correctness metrics and preference metrics. With these vibes, we achieve a model matching accuracy of **71.29%** and a preference prediction accuracy of **61.42%**.

---

**Prompt:** Summarize the article in 3 sentences.

> **Command X:** A giant filter-feeding 'lobster' took the place of whales 480 million years ago, a new fossil find has shown. The 6.5ft (two-metre) prehistoric creature was an early ancestors of modern crustaceans, insects and spiders. At this length it would have been comparable in size to athlete Usain Bolt. The remains were unearthed in Morocco and it is thought to be the oldest giant filter-feeder ever discovered.

> **TNLGv2:** A giant filter-feeding 'lobster' took the place of whales 480 million years ago . 6.5ft (two-metre) prehistoric creature belonged to the family of anomalocaridids . It was a 'gentle giant' .

---

## 6.2 HOW DO DIFFERENT LLMS SOLVE MATH PROBLEMS?

Objective tasks like math have a single final answer, but the way a model explains its thought process varies across models. We run VibeCheck on the MATH dataset ([Hendrycks et al., 2021c](#)) using chain-of-thought prompting to discover how GPT-4o and Llama-405b differ in their thought process and presentation. To reduce the variance seen from incorrect examples, we run VibeCheck only on the questions where both models answered correctly and aim to discover why GPT-4o is favored in 76% of conversations. Inspecting the top 5 vibes in [Figure 4](#), we observe that Llama-405b organizes its responses under markdown headings, adopts a more conversational tone, and includes overly detailed step-by-step explanations, as illustrated below. Examining the coefficients of the preference prediction model, we find that a formal tone and frequent use of notation positively correlate with preference, while over-explaining the reasoning process negatively correlates with preference. These vibes achieve a model-matching accuracy of **97.09%** and a preference prediction accuracy of **72.79%**.

---

**Prompt:** Find the positive difference between 2(6) + 4(3) and 8(3+3)

> **GPT-4o:**
> To find the positive difference between 2(6) + 4(3) and 8(3+3), we first need to evaluate each expression separately.
>
> First, evaluate 2(6) + 4(3)
>
> $2(6) + 4(3) = 2 \cdot 6 + 4 \cdot 3 = 12 + 12 = 24$
>
> Next, evaluate 8(3+3):
>
> $8(3 + 3) = 8 \cdot (3 + 3) = 8 \cdot 6 = 48$
> ...

> **Llama-405b:**
> ## Step 1: Evaluate the expression 2(6) + 4(3)
> First, we need to evaluate the expression 2(6) + 4(3) by following the order of operations (PEMDAS/BODMAS).
> We start by multiplying 2 and 6, which gives us 12. Then, we multiply 4 and 3, which gives us 12.
> Finally, we add the two results together, giving us 12 + 12 = 24.
>
> ## Step 2: Evaluate the expression 8(3+3)
> Next, we need to evaluate the expression 8(3+3) by following the order of operations (PEMDAS/BODMAS). We start by evaluating the expression inside the parentheses, which gives us 3 + 3 = 6. Then, we multiply 8 by 6, which gives us 48.
> ...

---

| **Vibe** (low -> high) | **Sep Score** [-0.9,0.9] | **PP Coef** [-0.6,0.6] | **Cohen** |
|---|---|---|---|
| **Explanation and Step-by-Step Detail.** Continuous narrative, no explicit step labels, less granularity. -> Detailed steps, labels (e.g., '## Step 1'), granular breakdown. | ▬ | ▬ | 0.90 |
| **Structural Formatting.** Continuous prose without explicit structuring. -> Uses headings, subheadings, numbered lists. | ▬ | ▪ | 0.70 |
| **Conciseness.** Extended discussions, unnecessary commentary, contains repetition. -> Brief, to-the-point explanations, no unnecessary repetition. | ▪ | ▪ | 0.51 |
| **Efficiency of Steps.** Detailed intermediary steps, broader explanations. -> Concise, straightforward solution steps. | ▪ | ▬ | 0.42 |
| **Mathematical Notation Use.** More written explanations, fewer symbols. -> Frequent use of symbols, LaTeX/MathML formatting. | ▪ | ▬ | 0.33 |

Figure 4: **Top 5 vibes comparing GPT-4o to Llama-3-405B on MATH CoT.** Negative separability scores indicate GPT-4o aligns with the low (red) description, while negative preference coefficients show alignment with low descriptions is preferred. GPT-4o outputs contain more LaTex/MathML formatting which positively correlated with human preference while Llama-3-405B has very structured and overly-detailed responses, which is negatively correlated with preference.

## 6.3 WHAT ARE VLM'S CAPTIONING STYLE?

Image captioning is one of the most popular use cases for Vision and Language models, but different captioning models focus on different image properties. We run VibeCheck on captions generated by GPT-4V ([Chen et al., 2023](#); [OpenAI, 2023](#)) and Gemini-1.5-Flash ([Reid et al., 2024](#)) on 1000 COCO images and we find that GPT-4V uses more poetic language and structures its captions as a dynamic

story, inferring the personality and emotions of the subjects in the image while Gemini sticks to more literal descriptions (Figure 16). The top 10 vibes generated by VibeCheck are able to achieve near perfect **99.13%** model matching accuracy and **89.02%** preference prediction accuracy. Although we compared the captions without the image in this experiment due to cost, the VibeCheck framework can be easily adapted to the multimodal setting.

**Prompt:** Describe the following image.

> **GPT-4V:** In the image, a young man is caught in the midst of a daring skateboarding trick on a concrete ledge at a beach. His black t-shirt, adorned with a white logo, contrasts with his black shorts and vibrant red shoes. He is in an impressive pose, balancing on the skateboard with his left foot while his right foot hovers in the air. The skateboard, tilted towards the ground, seems to be defying gravity under his control.
>
> The beach forms a lively backdrop to this scene. People can be seen enjoying their time, adding to the dynamic atmosphere. A red and white striped awning stands out, adding a splash of color to the scene. The image captures a moment of thrill and skill, set against the relaxed ambiance of a beach day.

> **Gemini-1.5-Flash:** The image shows a young man skateboarding in a skatepark. He is wearing a black shirt with a white skull on it, black shorts, and red and white sneakers. He is in mid-air, with his skateboard underneath him. The background is a beach, and there are people in the distance. The man is focused on his skateboarding, and he appears to be having fun.

**Creative Writing: GPT-4V:** Contains narrative embellishments and subjective interpretations. **Gemini-1.5-Flash:** Straightforward reporting without creative embellishments
**Color and Atmosphere Description: GPT-4V:** Emphasizes mood and overall ambiance, including colors and lighting. **Gemini-1.5-Flash:** Focuses on identifying and detailing objects without describing the ambiance.
**Descriptive Richness: GPT-4V:** Rich, detailed scene description with story element. **Gemini-1.5-Flash:** Simple, factual description without additional details.

## 7 LIMITATIONS

Although VibeCheck quantifies the impact of each vibe on model identity and user preference, it is challenging to disentangle whether a specific vibe directly influences human preference or if other confounding factors are at play. For example, a model might exhibit a vibe of being more engaging, but its preference by users could stem from its factual accuracy, where accurate outputs incidentally appear more engaging due to their clarity or relevance. Furthermore, the LLM-based vibe discovery process may not capture all relevant differences between models. This is particularly problematic when there's a significant discrepancy in model accuracy, as the discovered vibes may focus primarily on accuracy-related aspects. VibeCheck is also costly to validate, as each judge will have to evaluate each sample in $D_{validation}$ on each vibe. In order for this to be feasible, our method uses relatively inexpensive models such as GPT-4o-mini, but these judge models are often incorrect in their predictions, as shown in Figure 5. LLM judges also have biases (Zheng et al., 2023), like favoring their own outputs, which may affect the scoring. Lastly, running VibeCheck multiple times can lead to different vibes and different results, making it harder to reproduce findings exactly.

## 8 CONCLUSION

It may seem unconventional to focus on vibes instead of concrete metrics of correctness, but these qualitative properties have a measurable impact on how people judge models. VibeCheck provides a valuable addition to existing metrics for correctness by capturing these qualitative aspects that influence human preference. As LLM usage expands, we anticipate an increased focus on evaluating vibes to better align with user preferences. Moreover, this approach can be extended to other modalities, such as audio or visual content, and can be applied to compare any pairwise set of texts, making it a versatile tool for model evaluation. In future work, we hope to explore extending this framework to compare a larger number of models along with developing interventions which can use these vibes to improve human preference for given models.

**Acknowledgments.** We thank Ruiqi Zhong for introducing us to the joys of automated data analysis and Ion Stoica for insightful rants on evaluations beyond accuracy, as well as their feedback on the manuscript. We also thank Wei-Lin Chiang, Evan Frick, Tianle Li, and Issac Ong for co-authoring a blog post on the behaviors of Llama-3, which inspired one of the coolest experiments in this paper. Lastly, Lisa personally extends her appreciation to Joey, Jacob, and Trevor for embracing the writing of a paper that unironically uses the word "vibe" over 290 times. This paper has spawned many amusing quotes, such as: "Can we put confidence intervals on vibes?", "What if we call it 'No Numbers Just Vibes', and we replace all numbers with emojis?", and of course "I'm all vibed-out".

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

## A  EXPERIMENTAL DETAILS & DATASET STATISTICS

| Dataset | # Train | # Test |
|---|---|---|
| Human VS ChatGPT | 250 | 250 |
| Chatbot Arena - All | 839 | 839 |
| Chatbot Arena - STEM | 346 | 347 |
| Chatbot Arena - Writing | 278 | 277 |
| CNN/DailyMail | 444 | 346 |
| MATH | 218 | 218 |
| COCO w/ ShareGPT-4V Captions | 323 | 346 |

Table 3: **Dataset Statistics**

| Dataset | Model A | Model B | Model A Win Rate |
|---|---|---|---|
| Human VS ChatGPT | Humans | GPT-3.5 | - |
| Chatbot Arena - All | Llama3-70b-Instruct | GPT-4 + Claude-3-Opus | 50% |
| Chatbot Arena - STEM | Llama3-70b-Instruct | GPT-4 + Claude-3-Opus | 44% |
| Chatbot Arena - Writing | Llama3-70b-Instruct | GPT-4 + Claude-3-Opus | 57% |
| CNN/DailyMail | Cohere Command X | TNLGv2 | 71.12% |
| MATH | GPT-4o | Llama3-405b | 76% |
| COCO w/ ShareGPT-4V Captions | GPT-4V | Gemini-1.5-Flash | 80% |

Table 4: **Model Win Rates**

| Dataset | $d$ | $batch$ | $num\_eval\_vibes$ | $num\_final\_vibes$ | $iterations$ |
|---|---|---|---|---|---|
| Human VS ChatGPT | 40 | 5 | 10 | 10 | 3 |
| Chatbot Arena - All | 20 | 5 | 10 | 10 | 3 |
| Chatbot Arena - STEM | 20 | 5 | 10 | 10 | 3 |
| Chatbot Arena - Writing | 20 | 5 | 10 | 10 | 3 |
| CNN/DailyMail | 20 | 2 | 10 | 10 | 3 |
| MATH | 20 | 5 | 10 | 10 | 1 |
| COCO | 20 | 5 | 10 | 10 | 1 |

Table 5: **VibeCheck Hyperparameters**

$num\_eval\_vibes$ = number of vibes to validate at every iteration

$d$ = number of prompt output triples to use in each iteration of the vibe discovery phase

$batch$ = number of triples to feed into the prompt of the discovery LLM at once.

$iterations$ = number of vibe iterations to perform

$num\_final\_vibes$ = number of vibes to evaluate at the end of all the iterations. This can be set to false, in which case all the vibes collected in the iteration

We take the 1000 captions generated by GPT-4V from the ShareGPT-4V dataset Chen et al. (2023) and generate captions for the same images using the same captioning prompt using Gemini-1.5-Flash.

# B  GOLD STANDARD LABELS

Below is a summary of key differences found by human evaluators in the HC3 dataset Guo et al. (2023) listed in their paper.

**Characteristics of ChatGPT**

(a) Responses are well-organized, often starting with a definition of key concepts before providing a step-by-step explanation and concluding with a summary.

(b) Answers tend to be detailed and extensive.

(c) ChatGPT generally minimizes bias and avoids generating harmful content.

(d) It refrains from responding to queries beyond its scope of knowledge.

(e) In some cases, it may generate incorrect or fabricated information.

**Differences Between Human and ChatGPT Responses**

(a) ChatGPT remains strictly on topic, while human responses may shift toward related or tangential subjects.

(b) It tends to provide objective, fact-based answers, whereas human responses often include personal opinions or subjective elements.

(c) ChatGPT's tone is typically formal and structured, while human speech is more conversational and informal.

(d) Unlike humans, ChatGPT does not express emotions, relying solely on linguistic structure rather than emotional cues like punctuation or tone variations.

# C  GENERATING PRESET VIBES

| Vibe | Axis Definition (low → high) |
|---|---|
| Assertiveness | Uses tentative or uncertain language. → Uses definitive, confident statements. |
| Detail & Elaboration | Gives brief or shallow responses. → Provides thorough, nuanced, and expansive information. |
| Formality | casual, conversational, or informal language. → formal, sophisticated language and sentence structure. |
| Emotional Tone | Remains neutral or detached. → Infuses responses with expressive emotion and enthusiastic or empathetic tone. |
| Creativity & Originality | Sticks to standard, predictable answers. → Provides responses with novel ideas or imaginative scenarios. |
| Explicitness | Uses vague or implicit language. → States things directly and unambiguously. |
| Humor and Playfulness | Responds in a straightforward and serious manner. → Uses humor, playful language, or wordplay. |
| Engagement | Presents information passively. → Actively engages the reader using rhetorical questions or interactive phrasing. |
| Logical Rigor | Provides conclusions without thorough justification. → Constructs well-supported arguments with clear reasoning. |
| Conciseness | Uses verbose language and excessive details. → Uses minimal words to convey a point clearly. |

Table 6: **Predefined vibes.** We prompt GPT-4o to generate a set of 10 vibes which represent common axes on which LLM outputs differ.

We generate our list of 10 preset vibes by prompting GPT-4o with the following:

---

**Preset Vibe Generation Prompt**

```
I am a machine learning researcher trying to figure out the major
differences between the behavior of different large language models.
Can you list common ways in which two language models can differ in
their outputs?

Please output a list differences between these sets of outputs with
relation to specific axes of variation.  Try to give axes that a human
could easily interpret and they could understand what it means to
be higher or lower on that specific axis.  Please ensure that the
concepts used to explain what is high and low on the axis are distinct
and mutually exclusive such that given any tuple of text outputs, a
human could easily and reliably determine which model is higher or
lower on that axis.

The format should be
- {axis 1}:  {difference}
- {axis 2}:  {difference}

Please output differences which have a possibility of showing up in
future unseen data and which would be useful for a human to know
about when deciding with LLM to use.  For each axis, define clearly
and succinctly what constitutes a high or low score, ensuring these
definitions are mutually exclusive.  Please give 10 differences
```

## D  ADDITIONAL VIBECHECK DETAILS

### D.1  VIBE DISCOVERY

Below is the user prompt we use for vibe discovery.

---

**Vibe Discovery Prompt**

```
The following are the results of asking a set language models to
generate an answer for the same questions:
[PROMPT] [OUTPUT 1] [OUTPUT 2]
I am a machine learning researcher trying to figure out the major
differences between these two LLM outputs so I can better compare the
behavior of these models.  Are there any variations you notice in the
outputs?
Please output a list differences between these sets of outputs with
relation to specific axes of variation.  Try to give axes that a human
could easily interpret and they could understand what it means to
be higher or lower on that specific axis.  Please ensure that the
concepts used to explain what is high and low on the axis are distinct
and mutually exclusive such that given any tuple of text outputs, a
human could easily and reliably determine which model is higher or
lower on that axis.
The format should be:  {{axis}}:  Low:  {{low description}}; High:
{{high description}}
```

**Vibe Summarization.** To summarize the set of vibes found in the vibe discovery process, We cluster the axes using agglomerative clustering on the embeddings of the axes generated by the 'hkunlp/instructor-xl' model, and prompt GPT-4o to reduce this set by removing any vibes which are similar. After this stage we are left with a set of less than 20 vibes which we use to score the outputs of each model.

---

**Vibe Reduction Prompt**

```
Below is a list of axes with a description of what makes a piece of
text low or high on this axis.  Are there any axes that have similar
meanings based off their low and high descriptions?  Are there any
sets of axes that would convey the same information to a user (e.g.
level of detail)?  Could any of the low and high descriptions be
simplified to make them easier to understand?
Please remove any axes with roughly the same meaning and simplify the
descriptions of what makes a piece of text low or high on this axis.
Please ensure that the descriptions of what makes a piece of text low
or high on this axis are distinct, useful, and mutually exclusive.
Given any piece of text, a human should be able to easily and reliably
determine if this text falls high or low on each axis.
Here is the list of axes:  {axes}

Please return the simplified list of axes and the descriptions of what
makes a piece of text low or high on this axis.  These axes should
contain only one concept and should be human interpretable.  Some
examples of bad axes include:
- "Configuration Clarity:  High:  Clearly defined structure and
purpose.  Low:  Vaguely defined, minimal purpose." -> This axes is
bad because it is not clear what a clearly defined purpose means nor
what a vaugely defined purpose means.
- "Language and Communication:  High:  Varied/precise, complex
structure.  Low:  Straightforward, simple or general language." ->
This axes is bad because it combines multiple concepts into one axis.
- "Content Quality:  High:  High quality, engaging, informative.  Low:
Low quality, unengaging, uninformative." -> This axes is bad because
it is not clear what high quality means nor what low quality means.

Some examples of good axes include:
- "Complexity:  High:  Complex, multi-layered, intricate.  Low:
Simple, straightforward, easy to understand."
- "Efficiency (coding):  High:  Code optimized for runtime, minimal
memory usage.  Low:  Code inefficient, high memory usage."

Some examples of axes which should be combined include:
- "Emotional Tone:  High:  Contains emotionally charged language.  Low:
Maintains a neutral tone." and "Empathy:  High:  Shows empathy.  Low:
Only factual answers without empathy." are redundant because they both
measure the emotional content of the text.  If two similar axes are
found, keep the one that is more informative or more specific.

Please maintain the format of the original axes and return a list like
["{axis name}:  High:  {high description} Low:  {low description}",
...].  I should be able to parse this output into a string using
ast.literal_eval.  If the original list does not contain any redundant
axes, please return the original list.
```

If the number of vibes after the first reduction step is $> K$, we prompt GPT-4o to reduce the set further with the final reducer prompt.

**Final Vibe Reducer Prompt**

```
Below is a list of axes with a description of what makes a piece of
text low or high on this axis.  I would like to summarize this list to
at most number representative axes.

Here is the list of axes:  [VIBES]

These axes should contain only one concept and should be human
interpretable.  Some examples of bad axes include:
- "Configuration Clarity:  High:  Clearly defined structure and
purpose.  Low:  Vaguely defined, minimal purpose." -> This axis is
bad because it is not clear what a clearly defined purpose means nor
what a vaguely defined purpose means.
- "Language and Communication:  High:  Varied/precise, complex
structure.  Low:  Straightforward, simple or general language." ->
This axis is bad because it combines multiple concepts into one axis.
- "Content Quality:  High:  High quality, engaging, informative.  Low:
Low quality, unengaging, uninformative." -> This axis is bad because
it is not clear what high quality means nor what low quality means.
Some examples of good axes include:
- "Complexity:  High:  Complex, multi-layered, intricate.  Low:
Simple, straightforward, easy to understand."
- "Efficiency (coding):  High:  Code optimized for runtime, minimal
memory usage.  Low:  Code inefficient, high memory usage."

Some examples of axes which should be combined include:
- "Emotional Tone:  High:  Contains emotionally charged language.  Low:
Maintains a neutral tone." and "Empathy:  High:  Shows empathy.  Low:
Only factual answers without empathy." are redundant because they both
measure the emotional content of the text.  If two similar axes are
found, keep the one that is more informative or more specific.
Please return the simplified list of <=[K] axes with any redundant
axes removed and the descriptions of what makes a piece of text low or
high on this axis simplified.  Are there any axes which convey roughly
the same information?  Are there any axes where almost all samples
which score highly on one axis would also score highly on the other?

Please maintain the format of the original axes and return a numbered
list.  Each element should be structured as follows:  "{axis name}:
High:  {high description} Low:  {low description}"
```

## D.2 VIBE VALIDATION

**Prompt for ranker judge**

```
I want to compare the outputs of two language models (A and B) for the
same prompt.  I would like you to evaluate where each output falls on
the following axis:  [VIBE].
If you had to choose which output is higher on the axis, which
would you choose?  Here is the prompt and the outputs of A and B
respectively:
[PROMPT][OUTPUT A][OUTPUT B]
Please respond with which model you think is higher on the axis and
explain your reasoning.  If this axis does not apply to these examples
or these outputs are roughly equal on this axis, return "N/A".
```

### D.3  VIBE ITERATION

At iteration step $t$, we are left with $k$ distinct vibes which are well-defined and differentiating along with their scores $\nu_{1:k}(p, o_A, o_B)$. Using these scores, we train a LR model to predict LLM identity (i.e. "Is the response shown first LLM A or LLM B?") and get the predictions on our entire set $D$. Assuming we have not hit the max iteration steps set by the user, we iterate if the number of samples misclassified by the model matching predictor is greater than the number of prompts to perform discovery on ($d$). In iteration step $t + 1$, we take these misclassified prompt output triples in batches of size $batch$ along with the current set of vibes $\nu_1, ..., \nu_k$ and prompt the LLM to generate new differences between outputs what are not represented in the current vibes. These vibes are then reduced using the same procedure as the vibe discovery process. In practice we found that often some of the reduced vibes from the discovery phase at $t + 1$ were redundant with an existing axis, so we preform one more deduplication step using the prompt below.

---

**Vibe Discovery Iteration step**

```
Given a new set of respenses, your task is to expand on the set of
axes which have been previously identified by finding other clear
differences between the responses that are not captured by the
existing axes.  The expanded axes should be any differences between
responses that are not clearly captured by the existing axes.  Be as
exhaustive as possible in listing differences on as many different
axes as you can think of, and be specific about what constitutes high
and low on each axis.

Your axis should be interpretable:  a human should easily and reliably
determine which response is higher, lower, or even on this axis
when given a new set of responses.  Please do not make your axes
too broad and list as many axes as you can think of that are not
covered by the existing axes.  Most of these new axes should be either
completely different from the existing axes or should highlight a
more finegrained difference which an existing axis might broadly
cover.  For instance, if an existing axis is "Enthusiasm:  High:
enthusiastic, Low:  unenthusiastic", a new axis might be "Use of
Exclamation Points", or if an existing axis is "Cultural Context:
High:  culturally relevant, Low:  culturally irrelevant", a new axis
might be "Use of Slang".  ", a new axis might be "Use of Exclamation
Points", or if an existing axis is "Context", a new axis might be "".

Please think through the axes carefully and make sure they are clear,
concise, and do not overlap with eachother or the existing axes.  Do
not include any of the existing axes in your response.  Your output
should be in this format:

New Axes:
- axis 1:
High:  description of high
Low:  description of low

- axis 2:
High:  description of high
Low:  description of low

Do not include any other information in your response.
```

---

**Vibe deduplication in iteration step $t + 1$**

```
Here is a list of axes on which two strings may vary.  Each axis has a
description of what makes a string high or low on that axis.

[EXISTING AXES]
[NEW AXES]

It is likely that several of these axes measure similar things.
Your task is to remove any redundant axes.  Think about if a user
would gain any new information from seeing both axes.  For example,
"Emotional Tone:  High:  Contains emotionally charged language.  Low:
Maintains a neutral tone." and "Empathy:  High:  Shows empathy.  Low:
Only factual answers without empathy." are redundant because they both
measure the emotional content of the text.  If two similar axes are
found, keep the one that is more informative.

Output the reduced list of axes, separated by a newline.  All of the
axes should maintain the same format they have in the list of {axis}:
High:  {high} Low:  {low}
```

---

## D.4 GENERATING PREFERENCE LABELS

**prompt for generating preference labels**

```
Please act as an impartial judge and evaluate the quality of the
responses provided by two AI assistants (A and B) to the user question
displayed below.  You should choose the assistant that follows the
user's instructions and answers the user's question better.  Your
evaluation should consider factors such as the helpfulness, relevance,
accuracy, depth, creativity, and level of detail of their responses.
Begin your evaluation by comparing the two responses and provide a
short explanation.  Avoid any position biases and ensure that the
order in which the responses were presented does not influence your
decision.  Do not allow the length of the responses to influence your
evaluation.  Do not favor certain names of the assistants.  Be as
objective as possible.

Here is the prompt and the outputs of A and B respectively:
[PROMPT][OUTPUT A][OUTPUT B]

Please respond with the model which contains a higher quality response.
Based on your analysis, please explain your reasoning before assigning
a score.  Use the following format for your response:
Analysis:  {reasoning}
Model:  {A, B, tie}
```

---

## E FURTHER RELATED WORKS

**Automatic metrics for benchmark evaluations.** The number of benchmarks in the NLP community has exploded in recent years, with a wealth of work on providing a more holistic evaluation of language models beyond just accuracy. Several works Pang et al. (2020); Banerjee & Lavie (2005); Sellam et al. (2020), aim to improve on automatic metrics like BLEU Papineni et al. (2002) and ROUGE Lin (2004) scores to better measure how well a models output aligns with the ground truth by incorporating more nuanced evaluation criteria like factual accuracy, fluency, and conciseness. Similarly, efforts have been made Liang et al. (2023) to standardize model evaluation by evaluating models on many of these metrics.

## F   LIMITATIONS

**Prompt:** The following equation describes a parabola. If "a" is negative, which way does the parabola open? x = ay^2

> **Llama:** A nice simple question! ... The correct answer is down.

> **GPT:** If \(a\) is negative, the parabola opens to the left.

**Technical Accuracy and Terminology:** Factually correct, precise terminology → Factual inaccuracies, vague terminology.
**Ranker Decision:** LLama **Correct Answer:** Left (GPT's answer)

Figure 5: **Weaknesses in the mathematical abilities of the LLM judge (GPT-4o-mini).**

**Prompt:** Is European Thyroid Journal only open access or mixed open access and subscription?

> **Llama:** ...the European Thyroid Journal (ETJ) is a hybrid open-access journal, which means it offers both open-access and subscription-based models.

> **GPT:** As of my last update in 2023, the European Thyroid Journal operates under an Open Access model.

**Vibe:** Technical Accuracy and Terminology: High: Factually correct with precise terminology. Low: Factual inaccuracies and vague terminology.

Figure 6: The answer to certain questions changes depending on the following parameters:
(1) When was the question asked?
(2) What is the knowledge cutoff of Model A and Model B?
(3) What is the knowledge cutoff of the LLM ranker ensemble?
These types of questions lead to unreliable ranker evaluations and reduced inter-annotator agreement.

## G   VIBES FROM EACH APPLICATION

| Vibe (low -> high) | Sep Score [-0.5,0.5] | PP Coef [-3.6,3.6] | Cohen |
|---|---|---|---|
| **Conciseness.** Elaborate and lengthy explanations. **->** Short and to the point. | | | 0.37 |
| **Citation and References.** Avoids citations, smoother text flow. **->** Includes references and citations for credibility. | | | 0.41 |
| **Emotional Tone and Empathy.** Clinical and straightforward, less emotional engagement. **->** Uses comforting language, acknowledges emotional challenges. | | | 0.46 |
| **Technical Depth.** Simplified, general, and basic technical explanations. **->** Detailed, formal, and multifaceted technical explanations. | | | 0.65 |
| **Legal and Safety Considerations.** Does not consistently include disclaimers. **->** Includes disclaimers or notes about advice limitations. | | | 0.29 |
| **Contextual Information.** Focuses strictly on the topic. **->** Provides additional irrelevant context and discussion. | | | 0.39 |
| **Practical Advice and Safety.** Addresses concerns directly, less emphasis on professional help. **->** Practical, cautious advice, emphasizes seeking professional help. | | | 0.43 |
| **Detail Orientation.** Concise and limited responses covering fewer aspects. **->** Thorough and comprehensive responses covering multiple aspects. | | | 0.55 |
| **Response Length.** Short, to-the-point responses. **->** Long, informative responses. | | | 0.50 |
| **Formality and Tone.** Casual, relaxed tone with conversational language. **->** Formal, academic tone throughout. | | | 0.64 |

Figure 7: Human VS ChatGPT outputs on HC3 (Guo et al., 2023)

| Vibe (low -> high) | Sep Score [-0.3,0.3] | PP Coef [-0.7,0.7] | Cohen |
|---|---|---|---|
| **Engagement.** Presents information passively. -> Actively engages the reader using rhetorical questions or interactive phrasing. | | | 0.48 |
| **Emotional Tone.** Remains neutral or detached. -> Infuses responses with expressive emotion, making the tone enthusiastic or empathetic. | | | 0.53 |
| **Humor and Playfulness.** Responds in a straightforward and serious manner. -> Uses humor, playful language, or wordplay to make the response engaging. | | | 0.64 |
| **Creativity and Originality.** Sticks to standard, predictable answers. -> Provides responses with novel ideas or imaginative scenarios. | | | 0.51 |
| **Detail and Elaboration.** Gives brief or shallow responses. -> Provides thorough, nuanced, and expansive information. | | | 0.60 |
| **Assertiveness.** Uses tentative or uncertain language. -> Uses definitive, confident statements. | | | 0.49 |
| **Explicitness.** Uses vague or implicit language. -> States things directly and unambiguously. | | | 0.43 |
| **Logical Rigor.** Provides conclusions without thorough justification. -> Constructs well-supported arguments with clear reasoning. | | | 0.48 |
| **Conciseness.** Uses verbose language and excessive details. -> Uses minimal words to convey a point clearly. | | | 0.40 |
| **Formalness.** Uses casual, conversational, or informal language. -> Uses formal and sophisticated vocabulary and sentence structure. | | | 0.50 |

Figure 8: Preset vibes on Chatbot Arena[Overall]

| Vibe (low -> high) | Sep Score [-0.4,0.4] | PP Coef [-0.5,0.5] | Cohen |
|---|---|---|---|
| **Language and Tone.** Professional, straightforward tone. -> Enthusiastic, friendly tone. | | | 0.51 |
| **Typographic Emphasis.** Minimal use of typographic emphasis, letting the text stand alone. -> Uses typographic emphasis like bold or italics to highlight key points. | | | 0.64 |
| **Interactivity.** Provides information passively without engaging the user. -> Encourages user interaction, such as posing questions or suggesting actions. | | | 0.44 |
| **Formatting Completeness.** Responses are minimally formatted, relying on plain text. -> Responses include comprehensive formatting, such as Markdown or additional stylistic elements. | | | 0.57 |
| **Examples and Illustrations.** Minimal examples. -> Provides multiple examples. | | | 0.61 |
| **Use of Humor.** Maintains a serious tone without humorous elements. -> Employs humor frequently to engage the reader. | | | 0.62 |
| **Use of Personal Pronouns.** Rarely or never uses personal pronouns. -> Frequently uses personal pronouns (I, we, you). | | | 0.32 |
| **Ethical Consideration.** Provides factual information without commenting on ethics. -> Offers ethical considerations in its responses. | | | 0.53 |
| **Humility.** Projects confidence and completeness without discussing limitations. -> Frequently acknowledges limitations in the response or areas of uncertainty. | | | 0.41 |
| **Formality Level.** Uses informal or conversational language. -> Uses formal language and expressions. | | | 0.45 |

Figure 9: VibeCheck vibes on Chatbot Arena[Overall]

| Vibe (low -> high) | Sep Score [-0.2,0.2] | PP Coef [-0.8,0.8] | Cohen |
|---|---|---|---|
| **Assertiveness.** Uses tentative or uncertain language. -> Uses definitive, confident statements. | | | 0.34 |
| **Conciseness.** Uses verbose language and excessive details. -> Uses minimal words to convey a point clearly. | | | 0.34 |
| **Creativity and Originality.** Sticks to standard, predictable answers. -> Provides responses with novel ideas or imaginative scenarios. | | | 0.47 |
| **Detail and Elaboration.** Gives brief or shallow responses. -> Provides thorough, nuanced, and expansive information. | | | 0.62 |
| **Emotional Tone.** Remains neutral or detached. -> Infuses responses with expressive emotion, making the tone enthusiastic or empathetic. | | | 0.45 |
| **Engagement.** Presents information passively. -> Actively engages the reader using rhetorical questions or interactive phrasing. | | | 0.35 |
| **Explicitness.** Uses vague or implicit language. -> States things directly and unambiguously. | | | 0.36 |
| **Formalness.** Uses casual, conversational, or informal language. -> Uses formal and sophisticated vocabulary and sentence structure. | | | 0.56 |
| **Humor and Playfulness.** Responds in a straightforward and serious manner. -> Uses humor, playful language, or wordplay to make the response engaging. | | | 0.59 |
| **Logical Rigor.** Provides conclusions without thorough justification. -> Constructs well-supported arguments with clear reasoning. | | | 0.45 |

Figure 10: Preset vibes on Chatbot Arena[STEM]

| Vibe (low -> high) | Sep Score [-0.3,0.3] | PP Coef [-0.5,0.5] | Cohen |
|---|---|---|---|
| **Engagement and Enthusiasm.** The response is more formal, neutral, and factual without engaging language. -> The response exudes enthusiasm and engages the reader, often employing exclamation points, a friendly tone, and casual conversational remarks. | | | 0.43 |
| **Error Handling.** Minimal or no error handling, assumes ideal scenarios. -> Includes comprehensive error handling and user input validation within the code. | | | 0.33 |
| **Handling of Uncertain Information.** States information definitively without disclaimers. -> Clearly indicates uncertainty or assumptions. | | | 0.38 |
| **Interactivity and Engagement.** Formal, direct tone focused on clarity. -> Engaging tone, tutorial-like. | | | 0.44 |
| **Jargon and Terminology.** Uses general language and avoids jargon. -> Uses specialized jargon and complex terms. | | | 0.37 |
| **Safety and Accuracy Emphasis.** Lacks explicit emphasis on safety or ethics. -> Includes disclaimers, emphasizes ethical considerations. | | | 0.26 |
| **Tone and Enthusiasm.** Neutral, utilitarian. -> Engaging, enthusiastic. | | | 0.44 |

Figure 11: VibeCheck vibes on Chatbot Arena [STEM]. Note that we only find 7 vibes which achieve a separability score on the training set about the 0.05 threshold.

| Vibe (low -> high) | Sep Score [-0.4,0.4] | PP Coef [-0.6,0.6] | Cohen |
|---|---|---|---|
| **Assertiveness.** Uses tentative or uncertain language. **->** Uses definitive, confident statements. | | | 0.56 |
| **Conciseness.** Uses verbose language and excessive details. **->** Uses minimal words to convey a point clearly. | | | 0.36 |
| **Creativity and Originality.** Sticks to standard, predictable answers. **->** Provides responses with novel ideas or imaginative scenarios. | | | 0.46 |
| **Detail and Elaboration.** Gives brief or shallow responses. **->** Provides thorough, nuanced, and expansive information. | | | 0.64 |
| **Emotional Tone.** Remains neutral or detached. **->** Infuses responses with expressive emotion, making the tone enthusiastic or empathetic. | | | 0.55 |
| **Engagement.** Presents information passively. **->** Actively engages the reader using rhetorical questions or interactive phrasing. | | | 0.55 |
| **Explicitness.** Uses vague or implicit language. **->** States things directly and unambiguously. | | | 0.41 |
| **Formalness.** Uses casual, conversational, or informal language. **->** Uses formal and sophisticated vocabulary and sentence structure. | | | 0.60 |
| **Humor and Playfulness.** Responds in a straightforward and serious manner. **->** Uses humor, playful language, or wordplay to make the response engaging. | | | 0.61 |
| **Logical Rigor.** Provides conclusions without thorough justification. **->** Constructs well-supported arguments with clear reasoning. | | | 0.45 |

Figure 12: Preset vibes on Chatbot Arena [Writing]

| Vibe (low -> high) | Sep Score [-0.4,0.4] | PP Coef [-0.7,0.7] | Cohen |
|---|---|---|---|
| **Humanness/Relatability.** Formal or technical language. **->** Relatable and human-like language. | | | 0.40 |
| **Emotion and Tone.** Remains neutral and monotonous. **->** Injects emotions and varies tone. | | | 0.53 |
| **Humor.** Remains serious or formal, with no attempt at humor even in suitable contexts. **->** Incorporates humor or light-hearted elements that enhance the response and fit the context. | | | 0.55 |
| **Narrative Creativity.** Predictable storylines. **->** Unique and imaginative ideas. | | | 0.46 |
| **Structural Organization.** Unorganized responses lacking clear structure. **->** Clearly structured responses with headings or lists. | | | 0.55 |
| **Empathy.** Detached and indifferent. **->** Deep understanding of emotions. | | | 0.53 |
| **Consistency of Persona.** Displays inconsistency in tone and style. **->** Maintains a consistent voice and style throughout. | | | 0.36 |
| **Ethical Nuance.** Offers black-and-white viewpoints. **->** Considers moral complexities. | | | 0.52 |
| **Formality.** Relies on informal, casual, or conversational language, with a relaxed or inconsistent tone. **->** Uses structured, professional, and polished language, maintaining formal tone throughout. | | | 0.55 |
| **Caution.** Offers bold or risky suggestions without considering potential drawbacks or limitations. **->** Provides careful, measured responses that consider potential risks or consequences, showing prudence. | | | 0.45 |

Figure 13: VibeCheck vibes on Chatbot Arena [Writing]

| Vibe (low -> high) | Sep Score [-0.4,0.4] | PP Coef [-2.3,2.3] | Cohen |
|---|---|---|---|
| **Clarity and Conciseness.** Detailed and sometimes overly descriptive, risking redundancy. -> Summaries are concise and clear with minimal details. | ■ | ■ | 0.43 |
| **Tone on Emotional Aspects.** Objective tone, factual summaries without emotion. -> Captures emotional aspects, includes quotes. | ■ | ■ | 0.44 |
| **Personal Details.** Omits personal details, summarizes key facts. -> Includes names and direct quotes of individuals. | ■ | \| | 0.42 |
| **Specificity of Examples.** Lacks concrete examples, speaks in generalities. -> Includes specific examples or anecdotes to illustrate points. | ■ | ■ | 0.45 |
| **Emphasis on Cause and Effect.** Focuses on event sequence, less clarity in causality. -> Highlights cause and effect relationships clearly. | ■ | ■ | 0.26 |
| **Coverage of Multiple Viewpoints.** Presents information from a single perspective. -> Discusses multiple perspectives or viewpoints. | ■ | ■ | 0.28 |
| **Introduction and Contextual Background.** Minimal or absent introduction; reads like bullet points. -> Provides broad context-setting or introductory sentences. | ■ | \| | 0.37 |
| **Contextual Emphasis.** Focuses narrowly on events and actions. -> Emphasizes broader societal elements and contexts. | ■ | ■ | 0.44 |
| **Depth of Explanation.** Offers surface-level explanations, lacks depth. -> Provides deep, thorough explanations. | ■ | ■ | 0.48 |
| **Conclusion Strength.** Ends abruptly or lacks conclusive statements. -> Clearly states outcomes or implications at the end. | ■ | ■ | 0.37 |

Figure 14: VibeCheck vibes comparing TNLGv2 to Command X Large Beta on CNN/DailyMail Summarization (Hermann et al., 2015).

| Vibe (low -> high) | Sep Score [-0.9,0.9] | PP Coef [-0.6,0.6] | Cohen |
|---|---|---|---|
| **Mathematical Notation Use.** More written explanations, fewer symbols. -> Frequent use of symbols, LaTeX/MathML formatting. | ■ | ■ | 0.33 |
| **Efficiency of Steps.** Detailed intermediary steps, broader explanations. -> Concise, straightforward solution steps. | ■ | ■ | 0.42 |
| **Conciseness.** Extended discussions, unnecessary commentary, contains repetition. -> Brief, to-the-point explanations, no unnecessary repetition. | ■ | ■ | 0.51 |
| **Structural Formatting.** Continuous prose without explicit structuring. -> Uses headings, subheadings, numbered lists. | ■ | ■ | 0.70 |
| **Explanation and Step-by-Step Detail.** Continuous narrative, no explicit step labels, less granularity. -> Detailed steps, labels (e.g., '## Step 1'), granular breakdown. | ■ | ■ | 0.90 |

Figure 15: VibeCheck vibes comparing GPT-4o to Llama-3-405B on MATH CoT (Hendrycks et al., 2021c). We only find 5 vibes because the vibe reduction step is not required to return ≤ 10 vibes and in this case found only 5 distinct vibes which are able to almost perfectly separate model outputs.

| Vibe (low -> high) | Sep Score [-1.0,1.0] | PP Coef [-15.6,15.6] | Cohen |
|---|---|---|---|
| **Detail Orientation.** Gives general or minimal descriptions; lacks specificity. **->** Provides intricate details; describes specific elements. | ▮ | \| | 0.32 |
| **Background Details.** Provides minimal or no information about the background. **->** Offers extensive context and description of surroundings. | ▮ | \| | 0.56 |
| **Emotion and Relationships.** Provides neutral descriptions without attributing feelings or relationships. **->** Infers emotions and relationships between subjects. | ▮ | \| | 0.60 |
| **Color and Atmosphere Description.** Focuses on identifying and detailing objects without describing the ambiance. **->** Emphasizes mood and overall ambiance, including colors and lighting. | ▮ | \| | 0.63 |
| **Sense of Space.** Focuses on concise, immediate area. **->** Describes spatial depth and broader setting. | ▮ | ▌ | 0.42 |
| **Focus on Context.** Focuses on enumerating visual elements without added context. **->** Provides added context or narrative elements. | ▮ | \| | 0.43 |
| **Contextual Storytelling.** Sticks to physical appearance without narrative. **->** Adds narrative context and interpretations. | ▮ | ▬ | 0.42 |
| **Artistic or Poetic Language.** Employs straightforward, factual, and plain language. **->** Uses vivid, imaginative, and expressive language. | ▮ | ▌ | 0.37 |
| **Descriptive Richness.** Simple, factual description without additional details. **->** Rich, detailed scene description with story elements. | ▮ | ▌ | 0.36 |
| **Creative Writing.** Straightforward reporting without creative embellishments. **->** Contains narrative embellishments and subjective interpretations. | ▮ | ▬ | 0.42 |

Figure 16: VibeCheck vibes comparing Gemini-1.5-Flash to GPT-4V on COCO Captions (Lin et al., 2014).

# H    MORE LLAMA VS GPT EXAMPLES

**Prompt:** If I was a mouse would we be on squeaking terms?

---

**Llama-3-70b:** Ahahaha! Oh, what a delightful pun! As a conversational AI, I'd say we're definitely on squeaking terms... or should I say, on paws-itive terms? Squeak squeak!

---

**GPT-4:** If you were a mouse, we'd find a way to communicate effectively, whether that involves squeaking, gestures, or any form of interaction that works for both of us. I'm here to assist, regardless of your form!

---

**Use of Humor:** **Llama-3-70b:** Employs humor frequently to engage the reader. **GPT-4:** Maintains a serious tone without humorous elements.

**Formality Level:** **Llama-3-70b:** Uses informal or conversational language. **GPT-4:** Uses formal language.

**Language and Tone:** **Llama-3-70b:** Enthusiastic, friendly tone. **GPT-4:** Professional, straightforward tone.

Figure 17: Squirrel example from Chatbot Arena

---

**Prompt:** what happens to olympic medals when atheltes are caught doping after the event?

---

**Llama-3-70b:** When an athlete is caught doping after an Olympic event, the International Olympic Committee (IOC) has a process in place to deal with the situation. Here's what typically happens:

1. **Investigation and Testing**: The IOC, in collaboration with the World Anti-Doping Agency (WADA) and the International Federation (IF) of the sport involved, conducts an investigation and testing to confirm the doping violation.
2. **Disqualification and Medal Reallocation**: If the doping violation is confirmed, the IOC disqualifies the athlete from the event and strips them of their medal. The IOC then reallocates the medals to the next eligible athletes in the competition.
3. **Return of Medal**: The disqualified athlete is required to return their medal to the IOC. If the athlete fails to return the medal, the IOC may take further action, including seeking legal recourse.
4. **Re-allocation Ceremony**: In some cases, a new medal ceremony may be held to award the medal to the new recipient(s).
5. **Record Correction**: The IOC updates the official Olympic records and results to reflect the changes.

Some notable examples:

* **Lance Armstrong (Cycling, 2000)**: Stripped of his bronze medal in the individual time trial after admitting to doping.
* **Marion Jones (Track and Field, 2000)**: Returned her three gold medals and two bronze medals after admitting to doping.
* **Russian Olympic Team (2014 Sochi Winter Olympics)**: Multiple athletes were disqualified and stripped of their medals due to a state-sponsored doping program.

It's worth noting that the IOC has a statute of limitations for doping violations, which is currently set at 10 years from the date of the Olympic Games. This means that if a doping violation is discovered after the 10-year period, the IOC may not be able to take action to strip the athlete of their medal.

The IOC's anti-doping efforts aim to protect the integrity of the Olympic Games and ensure a level playing field for all athletes.

---

**Claude-3-Opus:** When athletes are caught doping and disqualified after an Olympic event, the following typically happens to their medals:

1. The disqualified athlete must return their medal to the International Olympic Committee (IOC).

2. The IOC then reallocates the medals to the next eligible athletes based on the updated results.

3. The athletes who originally finished behind the disqualified competitor are awarded the appropriate medals.

- If the athlete who is disqualified won gold, the silver medalist is upgraded to gold, the bronze medalist is upgraded to silver, and the fourth-place finisher receives the bronze medal.
- If the disqualified athlete won silver or bronze, the medals are reallocated accordingly to the next finishers in the event.

4. In some cases, medal ceremonies are conducted to present the new medals to the rightful winners.

It's important to note that the process of disqualification, medal reallocation, and subsequent ceremonies can sometimes take years, depending on the timing of the doping revelation and the completion of legal proceedings or appeals.

---

**Examples:** **Llama-3-70b:** Broad range of detailed examples. **Claude-3-Opus:** Fewer and less detailed examples.

**Typographic Emphasis:** **Llama-3-70b:** Uses typographic emphasis like bold or italics to highlight key points. **Claude-3-Opus:** Minimal use of typographic emphasis, letting the text stand alone.

**Depth of Analysis:** **Llama-3-70b:** Detailed and in-depth analysis with numerous distinct points. **Claude-3-Opus:** Concise analysis with fewer, general points.

Figure 18: Olympics example from Chatbot Arena

**Prompt:** Is Iranian supreme leader irl Dr. Evil ?

---

**Llama-3-70b:** What a delightful question!

While Iranian Supreme Leader Ali Khamenei may be a controversial figure, I'm afraid he's not quite the same as Dr. Evil, the fictional villain from the Austin Powers movie series.

Here are a few reasons why:

1. **No cat-stroking**: Unlike Dr. Evil, Khamenei hasn't been known to stroke a cat menacingly while plotting world domination. He's a religious and political leader, not a Bond villain.
2. **No outlandish schemes**: While Khamenei has been involved in various political and military conflicts, there's no evidence of him concocting absurd plans like Dr. Evil's moon-based laser beam or sharks with frickin' laser beams attached to their heads.
3. **No comically exaggerated persona**: Khamenei is a serious political figure, not a caricatured villain with a penchant for dramatic entrances, over-the-top monologues, or outrageous fashion sense.
4. **No global domination ambitions**: While Iran's supreme leader may have regional and ideological ambitions, there's no indication he seeks to take over the world or hold it for ransom (unlike Dr. Evil's constant threats to destroy the world unless his demands are met).

In all seriousness, Ali Khamenei is a complex figure with a significant impact on Iranian politics and international relations. While his views and actions have been widely criticized, it's essential to approach discussions about him with nuance and respect for the complexity of the issues involved.

So, while Khamenei may not be Dr. Evil, I do hope this response has brought a smile to your face!

---

**GPT-4:** No, it would be inaccurate and inappropriate to equate Iran's Supreme Leader with a fictional villain like Dr. Evil from the Austin Powers movies. While Iran's government and leaders are controversial and face significant criticism internationally, making glib comparisons to over-the-top comedic characters trivializes the real-world complexities and sensitivities around this topic.

Iran's Supreme Leader is a religious and political position currently held by Ali Khamenei. The role wields significant power in Iran's theocratic system. There are many valid concerns and debates regarding Iran's human rights record, nuclear program, regional influence and tensions with other countries. However, these issues deserve to be discussed seriously with factual information, not facile pop culture references.

Whenever discussing foreign leaders and governments, especially those that are viewed negatively by many, it's important to be thoughtful and avoid hyperbole or caricature. Even when we disagree with a regime, we should strive to understand the historical context and represent the facts as objectively as possible. Demonizing others, even those we oppose, rarely leads to greater understanding or positive change.

---

**Ethical Considerations**: Llama-3-70b: Provides factual information without commenting on ethics.

Claude-3-Opus: Offers ethical consideration in its responses.

**Typographic Emphasis:** Llama-3-70b: Uses typographic emphasis like bold or italics to highlight key points.

Claude-3-Opus: Minimal use of typographic emphasis, letting the text stand alone.

**Depth of Analysis:** Llama-3-70b: Detailed and in-depth analysis with numerous distinct points.

Claude-3-Opus: Concise analysis with fewer, general points.

Figure 19: Supreme Leader example from Chatbot Arena

