# OpenReview forum: "VibeCheck: Discover and Quantify Qualitative Differences in Large Language Models"
_ICLR.cc/2025/Conference — ICLR 2025 Poster_

### Official Review · Reviewer_a334 · 2024-11-02

**Soundness:** 4
**Presentation:** 3
**Contribution:** 2
**Rating:** 6
**Confidence:** 4

**Summary:**

This paper aims to automate the vibes-based evaluation that is common among LLM practitioners. They first define three characteristics of a useful vibe and then use an LLM-aided process to summarize model output into different axes of vibes and iteratively refine them with model misclassification based on these models. They then conduct validation using both a data with gold answer as well as open-ended metrics. Afterwards, they apply VibeCheck to tasks including summarization, mathematics and image captioning.

**Strengths:**

This paper is well-motivated, as it basically provides an automatic way of conducting the “vibe-based” evaluation which is very commonly practiced nowadays, as the evaluation of LLMs have become difficult. The writing of the paper is very clear and easy to follow. The results presented are overall convincing. Also, it is very nice to review some HCI literature in the paper.

**Weaknesses:**

1) the author did not engage with the line of work that tries to characterize human preference into high-level principles (e.g. https://arxiv.org/abs/2406.06560, https://aclanthology.org/2024.acl-long.99/). It would be much appreciated if the authors could situate their work around these and articulate their contributions more clearly.
***
2) In Table 4, we see that the predefined vibes get to almost the same preference as VibeCheck. Thus, I am not entirely convinced whether we would need to discover these vibe aspects at run time, as something that could be done a-priori is already quite good, also given the compute overhead as mentioned? Perhaps you could convince me if you show that individual user considers drastically different vibes or the vibes considered for different application are extremely different.
***
3) More analysis on model matching and preference predicting accuracy would be great – why are these results so drastically different for each task? Does this also point to a limitation of the applicability of your proposed method, the fact that the "explained variance" of human preference by vibes are so different across tasks?
***
4) I think it is important to note somewhere in the paper that these vibes (and the way you conduct experiment) are post-hoc explanations that are correlated with user preference. However, they are not necessarily the reason that a user actually prefers one option over the other in real time. In other words, one limitation is that your study lacks ecological validity. However, I think this is ok in this context, as long as you acknowledge it. Of course, if you could show that these vibes are also things the users actually consider when they indicate their preference real-time, I will also be convinced.

**Questions:**

1) Could you inspect the cases that, despite all these vibes being generally predictive, the model matching prediction or the preference predictions are wrong? Could you try to “VibeCheck” those cases?

2) In the CNN/DailyMail experiment, the preference prediction accuracy is as low as 61.42%, which is not a whole lot better than random. Why should I still believe, then, that the vibes discovered in this case are “real” and actually useful?

---

> ### Author Response · Authors · 2024-11-26
> **An HCI person!!!!!**
>
> **Connection to other HCI works.** We are glad the reviewer appreciated our related works in HCI; we think it is an important area that often doesn’t get enough attention and appreciate the additional works you provided. Admittedly we were not aware of these before but agree that they are relevant. While both of the works you mentioned do target a similar goal as ours - finding factors which influence human preference - this is not our only goal. The reason we automatically discover axes and optimize for model matching accuracy is that we are equally interested in finding the axes on which models differ as we are the axes which correlate the most with human preference.
>
> Petridis et al. proposes more of an explainable recommender system which relies on more detailed human feedback like rewriting and critiquing, whereas VibeCheck only assumes user preference labels. Li et al’s work is more aligned with our work from a methodology side, although this method targets using preset vibes and grading on a likert scale. In contrast, our work aims to automatically find these axes and focuses on a comparative setting, which allows for a more fine-grained analysis of a pair of models rather than a more general analysis over a set of models. We will update our related works with this discussion and are happy to expand on any of these points if you find it insufficient.
>
> **Preset vibes are already good.** See global response.
>
> **Lack of ecological validity.** We agree that this is an important limitation of our method. We have added the following to our limitations section.
>
> “Although VibeCheck quantifies the impact of each vibe on model identity and user preference, it is challenging to disentangle whether a specific vibe directly influences human preference or if other confounding factors are at play. For example, a model might exhibit a vibe of being more engaging, but its preference by users could stem from its factual accuracy, where accurate outputs incidentally appear more engaging due to their clarity or relevance.”
>
> **Could you inspect cases where the MM and PP are wrong?** Yes! In fact this is what our iteration step does: it takes the samples which were misclassified by the model matching classifier train on the current list of vibes and uses these incorrect samples when proposing new vibes.
>
> **Why should I believe the vibes of the CCN/DailyMail experiment if the preference prediction is so low?** This is a great question, and admittedly one that we do not have a bulletproof answer to. We do have results in our global response that shows our vibes achieve a higher preference prediction than preset vibes, but it is unclear if the low preference prediction accuracy is because this preference is uninterpretable or because VibeCheck was not able to discover the relevant axes. Please let us know if you have other suggestions on how to best validate this and we are happy to try it out :)

---

> > ### Author Response · Authors · 2024-12-03
> > **Difference in MM and PP accuracy across tasks**
> >
> > We apologize for forgetting to address this!
> >
> > It is true that model matching (MM) scores and preference prediction (PP) scores vary by task. Looking at the results on Chatbot Arena for example, MM and PP scores are higher for writing tasks compared to STEM tasks. When examining the separability scores for preset vibes, we found that vibes related to assertiveness, emotional tone, and creativity are much lower for STEM tasks, while vibes like logical rigor show slightly higher separability scores. We suspect that many subjective stylistic elements are more prominent in open-ended tasks like writing, whereas STEM tasks are more objective. This difference likely makes it harder to predict model identity and user preference in the STEM category.
> >
> > Interestingly, during the initial VibeCheck runs on STEM tasks, many vibes were closely related to accuracy (e.g., “code correctness,” “accuracy of content”), suggesting it identified axes highly relevant to STEM. However, we deliberately adjusted our prompts to exclude these accuracy-related axes, as we aimed to focus on vibes unrelated to factual correctness. With this in mind, we believe that human preference in STEM tasks is likely less influenced by stylistic factors, and that the amount of explained variance in human preference generally differs across tasks. That said, we think it is likely that there are additional interpretable axes influencing human preference that VibeCheck did not find.
> >
> > | Vibe (low -> high)                                                                                                                                      |   Sep Score STEM |   Sep Score Writing |
> > |:--------------------------------------------------------------------------------------------------------------------------------------------------------|-----------------:|--------------------:|
> > | Creativity and Originality. Sticks to standard, predictable answers. $\to$ Provides responses with novel ideas or imaginative scenarios.                |             0.03 |                0.23 |
> > | Humor and Playfulness. Responds in a straightforward and serious manner. $\to$ Uses humor, playful language, or wordplay to make the response engaging. |             0.18 |                0.34 |
> > | Emotional Tone. Remains neutral or detached. $\to$ Infuses responses with expressive emotion, making the tone enthusiastic or empathetic.               |             0.2  |                0.32 |
> > | Assertiveness. Uses tentative or uncertain language. $\to$ Uses definitive, confident statements.                                                       |            -0.01 |                0.14 |

---

### Official Review · Reviewer_513D · 2024-11-03

**Soundness:** 3
**Presentation:** 3
**Contribution:** 3
**Rating:** 6
**Confidence:** 3

**Summary:**

LLM-generated responses are mostly evaluated on their correctness, dismissing other important dimensions such as its tone or other subjective dimensions.

Distinctive traits of an LLM are called _vibes_ in this paper. The paper introduces an approach, VibeCheck, to finding and measuring such vibes. It consists of an LLM that qualitatively compares a pair of models to find distinctive vibes between the two. It identifies which dimensions the generated outputs of the models differ on to get a set of vibes. LLM judges are then used to determine where the model's output falls on the vibe.

This is evaluated on three fronts:
1. well-defined (agreement among multiple users)
2. differentiating (ability. to distinguish between LLMs)
3. user-aligned (predictive of user preferences)

The authors conduct multiple experiments to evaluate the method.

**Strengths:**

1. The problem this paper addresses is very important and the approach is creative.
2. Table 1 shows the utility of the paper, it shows how similar the vibes are that VibeCheck finds to the ones that humans find.
3. The authors conduct extensive experiments to show how it performs on different datasets.
4. The paper is well-written and easy to follow.

**Weaknesses:**

1. A batch is used to find the vibes, however, there is not much discussion about strategies to select this batch. If the batch contains similar prompts and is low in diversity, then not all vibes might be considered. E.g. for vibes on STEM prompts to work, it will need to see something similar in the vibe batch, however, it is not ensured that this will be the case.
2. The LLMs used are still on the larger side (GPT4o-mini).

**Questions:**

In the vibe iteration process, how many more vibes were found?

---

> ### Author Response · Authors · 2024-11-26
> **We are finally getting "batch" to you**
>
> **Selection strategies for batch selection.** This is a fantastic question! To clarify, the batching stage is used only for vibe discovery. Once a vibe is identified, we evaluate each prompt-response triplet individually during vibe scoring. For the final version, we randomly sampled responses for each batch. However, we also experimented with clustering the prompts by topic and sampling within each topic. We ultimately did not include this method because the discovered vibes were similar to those found with random sampling. That said, this approach could be advantageous in scenarios where the prompt pool is highly diverse. We suspect that random sampling performs well because the context length is small enough for the proposal model to effectively enumerate differences both across responses and for each individual prompt. If time permits we will try to get the numbers on the effects of clustering prompts before sampling.
>
> **Vibes found per iteration.** We observe that most vibes are discovered during the first iteration. Using our hyperparameters, each iteration typically produces around 20 distinct vibes, but 10–15 of these are usually filtered out because they are similar to vibes from past iterations or they have a low separability score.  For example, in our chatbot arena experiments, where we restrict the final output to 10 vibes, 7 out of 10 vibes were identified in the first iteration.

---

### Official Review · Reviewer_HgoV · 2024-11-04

**Soundness:** 3
**Presentation:** 3
**Contribution:** 2
**Rating:** 6
**Confidence:** 4

**Summary:**

The paper introduces VibeCheck, an evaluation pipeline to identify and compare the qualitative characteristics ("vibes") of LLMs. They employed three stages - (1) vibe discovery using thoroughly designed prompts; (2) vibe validation; and (3) vibe iterations. They also propose a method to quantify the usefulness of VibeCheck by calculating several metrics such as agreement, model matching accuracy, and preference prediction scores. Then, the paper tested VibeCheck across various tasks such as summarization, math, and image captioning, and it revealed how different models exhibit unique behaviors across different domains.

**Strengths:**

- This work provides a novel ground to assess the subjective characteristics of LLM outputs that usually differ by task purposes. Not much work has focused on this angle of LLM evaluation.
- The paper attempted to provide a comprehensive framework to develop an automatic detection of "vibes" in different LLM outputs and how aligned these outputs are with human preferences.

**Weaknesses:**

- Overall, the paper is hard to read and understand. A large amount of effort is needed to restructure the organization of the paper.
- Too many typos and grammatical errors are found in both the paper and the appendix. Needs to be corrected.
- Results seem too weak to claim the usefulness of VibeCheck. For example, in Table 4, VibeCheck (even 3 runs) does not perform better than the pre-defined baseline, increasing doubt about whether this framework does better than a naive GPT4o model.

**Questions:**

- Sec 3: it could strengthen your claim of selecting those three evaluation criteria based on solid literature work. At least, it could be better to provide why these three criteria are important in measuring such qualitative behaviors of LLMs.
- L220-225: what are the criteria for those threshold setups (0.2, 0.05)?
- Please provide a better visualization of Table 3. It is hard to interpret the scores and their relationship with llama.
- Appendix B: as mentioned these are direct quotes from the original HC3 paper. Please put a citation of this paper; otherwise, it could be regarded as flagging behavior for ethics review.
- Please combine the main text and appendix into one.

---

> ### Author Response · Authors · 2024-11-26
> **We apologize for all the typos...**
>
> **The writing is hard to follow.** We apologize that our paper is hard to follow and corrected our many typos in the revised manuscript. However, we would like to point out that reviewers a334 and 513D found the paper well written and easy to follow. Please let us know if there are any specific parts of the method or results that are confusing and we are happy to explain!
>
> **Comparison to preset vibes.** Please refer to our global response.
>
> **Connect criteria to existing literature.** Our choice of the three evaluation criteria (well-defined, differentiating, and user-aligned) stems from fundamental principles in qualitative evaluation[1,2,3] - well-defined ensures reliability through consistent measurement, differentiating ensures the metric captures meaningful distinctions between models, and user-aligned validates that these distinctions matter in practice. These criteria parallel established frameworks for evaluating metrics in other fields like machine translation and dialogue systems, where metrics must demonstrate consistency, discriminative power, and correlation with human judgments[2,3].
>
> [1] Golafshani, Nahid. (2003). “Understanding reliability and validity in qualitative research.” Qual. Rep.. 8. 597-606.
> [2] Banerjee & Lavie, “METEOR: An Automatic Metric for MT Evaluation with Improved Correlation with Human Judgments”, ACL 2005
> [3] Mehri & Eskenazi, “Unsupervised Evaluation of Interactive Dialog with DialoGPT”, SIGdial 2020
>
>
> **Explanation of thresholds.** We selected 0.05 because if this difference is seen in less than 5% of samples it is likely not a distinguishing factor. We selected 0.2 as this indicates poor inter-annotator agreement [4].
>
> [4] McHugh ML. Interrater reliability: the kappa statistic. Biochem Med (Zagreb). 2012
>
> **Table 3 is hard to read.** We agree and have provided an updated visualization of Table 3 (now Figure 2) in the manuscript which displays the scores as bars to better illustrate the magnitude of the coefficients. Please let us know if this is easier to interpret!
>
> **Put HC3 citation in the appendix.** Thank you for pointing this out, we have added the citation and made it more apparent that this is an exact quote by putting the text in a colored box.

---

> > ### Comment · Reviewer_HgoV · 2024-12-02
> >
> > Thank you for your detailed responses. I have adjusted my scores.

---

> > > ### Author Response · Authors · 2024-12-02
> > > **thank you!**
> > >
> > > We are glad we have addressed your concerns and thank you for your increased score!

---

### Official Review · Reviewer_CAD4 · 2024-11-12

**Soundness:** 2
**Presentation:** 3
**Contribution:** 2
**Rating:** 3
**Confidence:** 4

**Summary:**

The paper’s goal is to discover LLM vibes, i.e. traits of a model which are well-defined, differentiating and user-aligned. The main contribution of the paper is a method (VibeCheck) for quantifying vibes of LLM models. The method first employs a LLM model to extract a set of vibes, then decide which model outputs are conforming to that respective vibe based on a panel of LLM judges. The authors validate to what extent their method aligns with human responses and user preferences in the context of three different tasks: text summarization, math problem solving and image captioning. This approach allows them to quantify vibes of popular LLM models, finding that Llama3 outputs tend to be more friendly in nature compared to outputs from GPT-4 and Claude, while the latter two models tend to be more formal in nature.

**Strengths:**

The paper is making an important observation that current LLM evaluations are mainly focusing on one textual dimension (for eg., correctness) at the expense of other aspects that matter for the end user, failing to capture the open-ended nature of LLM applications, nuanced qualitative aspects and their dependence on subjective user preferences; LLM evaluation is indeed an open and important research topic

The authors propose a definition of vibes and discuss what makes a vide valid accounting for aspects such as well-defined, differentiating, then validate the identified vibes against human gold labels and aim to quantify the differences between popular LLM models using their method

**Weaknesses:**

“Vibes” are subjective and not a well-defined concept; the paper defines vibes as identifying traits of LLM models and in contrast to concrete evaluation measures which measure objective aspects of the text, evaluating subjectivity is less clear and straightforward

The authors consider a vibe to be well-defined if multiple LLM judges agree, however it is possible that those LLM judges all have the same limitations; it is also unclear to what extent LLM judges decisions on what constitutes a valid vibe aligns with humans

The paper acknowledges that vibes are user and task dependent, however there is no accounting for these aspects in the experiments

The paper would benefit from more structure, more in-depth analysis and more detailed figure/table captions. The extra space could be used for including a discussion section to summarize all the findings and takeaways.

**Questions:**

Your definition of vibes is similar to how concept axes are defined in interpretability, have you tried connecting the two?

Why discover vibes and validate on gold labels when you can actually use the gold labels directly?

Have you considered the impact of the prompts you are using on eliciting specific model vibes?

Figure 1: It is not clear from the example provided for “What is a vibe?” that Output A differs from Output B in the friendliness direction (formal → friendly). Which answer is more formal and which answer is more friendly? Also why Vi() takes values of 1 and -1 when the prompt for the judge is asking the model to answer with A, B or equal? Figure 1 caption needs more explanation.

Table 4 - is the top table for overall results? Please indicate what the abbreviations mean - Model Matching (MM), Preference Prediction (PP)

Line 336 - typo “sensitive”

---

> ### Author Response · Authors · 2024-11-26
> **Thanks to you we have better figure captions now!**
>
> **Vibe Alignment with humans and gold labels.** We agree that what an LLM judge considers as a vibe is not guaranteed to align with what humans would think is a vibe. We aimed to address this question through our gold label comparison, which aims to validate that VibeCheck is aligned with what is currently done (humans looking at the data). Otherwise, there is a possibility that the vibes we discover are either (1) untrue but receive a high score due to some unknown bias in LLMs, or (2) technically correct but uninteresting or uninterpretable. We see in this evaluation that our LLM discovered vibes do indeed match the vibes found by human-based discovery.
>
> **User and task dependence.** Please refer to our global response.
>
> **Paper structure.** We appreciate this feedback and agree that more analysis of results would benefit the paper. We have updated the manuscript with more analysis, figure captions, and improved results visualizations. Please let us know if you have any other concerns surrounding the writing.
>
> **Clarification on Figure 1.** For the “what is a vibe” section, output A is considered more friendly, indicated by the “What a bold question!”. For vibe scoring, we parse the A/B/equal into 1/-1/0 as described in section 4. We have updated our figure and caption to provide a better explanation.
>
> **Table 4.** Yes the top table is overall results. We have condensed this table with the bottom table and clarified abbreviations in the new captions, along with the takeaways of the table.
>
> **Connection to concept axes.** Our definition of vibes is very similar to concept axes in interpretability, but used in a different context. Concept axes focus on attributing a model's latent representations to a human interpretable concept. Since vibes are also defined as a human interpretable property seen in model outputs, these could be seen as concept axes, although the concepts we see in interpretability literature usually take the form of more fine grained attributes like “state capitols” and “faces”[1,2,3,4] or operations like copy/paste[5]. That being said, we will gladly put more discussion on the connection to interpretability and welcome any other related works you think are relevant to compare to.
>
> [1] Gandelsman et al, “Interpreting CLIP’s Image Representation Via Text-Based Decomposition“, Proceedings of the 2024 International Conference on Learning Representations (ICLR 2024)
>
> [2] David Bau, Jun-Yan Zhu, Hendrik Strobelt, Agata Lapedriza, Bolei Zhou, and Antonio Torralba. Understanding the role of individual units in a deep neural network. Proceedings of the National Academy of Sciences (2020).
>
> [3] Koh et al, “Concept Bottleneck Models”, International Conference on Machine Learning (ICML 2020)
>
>
> [4] Ghorbani et al, “Towards Automatic Concept-based Explanations”, Neurips 2019
>
> [4] Eric Todd, Millicent L. Li, Arnab Sen Sharma, Aaron Mueller, Byron C. Wallace, and David Bau. "Function Vectors in Large Language Models." Proceedings of the 2024 International Conference on Learning Representations (ICLR 2024)

---

### Author Response · Authors · 2024-11-26
**An overdue reply to reviewers**

We apologize for our delayed response and understand if the reviewers are unable to fully engage before the discussion period ends.

We sincerely appreciate the quality of our reviewers’ feedback! We agree with most of your concerns and are grateful for the references to relevant HCI and interpretability literature, as well as the requests for additional analysis on how vibes and preferences vary across different users and tasks. We have corrected our (many) typos and have updated the paper with more thorough analysis of the results. Below we address some shared concerns across multiple reviewers:

## Proof that vibes are user and task dependent:

We demonstrate in the appendix that (1) the vibes identified for STEM tasks differ from those found in Writing tasks, and (2) user preferences for the same vibe differ between STEM and Writing tasks. Based on the reviewers’ feedback, we now realize this is a crucial experiment and will include it in the main paper.

### Task Dependence
While some vibes are shared across both question types, others are specific to the question type. Below are vibes which are unique to Writing and STEM questions. For instance, narrative creativity, consistency of persona, and humor are exclusive to Writing tasks, whereas error handling, a tutorial-like tone, and complex jargon are unique to STEM tasks.

| **Writing Vibes**                          | **STEM Vibes**                                |
|--------------------------------------------|-----------------------------------------------|
| Humor: Remains serious or formal, with no attempt at humor even in suitable contexts. → Incorporates humor or light-hearted elements that enhance the response and fit the context. | Interactivity and Engagement: Formal, direct tone focused on clarity. → Engaging tone, tutorial-like. |
| Narrative Creativity: Predictable storylines. → Unique and imaginative ideas. | Error Handling: Minimal or no error handling, assumes ideal scenarios. → Includes comprehensive error handling and user input validation within the code. |
| Consistency of Persona: Displays inconsistency in tone and style. → Maintains a consistent voice and style throughout.  | Jargon and Terminology: Uses general language and avoids jargon. → Uses specialized jargon and complex terms. | |

### Task and User Dependence
The table below presents the preference prediction scores for each vibe across writing and STEM tasks. A positive preference prediction (PP) coefficient indicates that responses aligned with the “high” description of a vibe are positively associated with preferred responses. The results highlight that detailed explanations, humor, and expressive emotion are positively correlated with human preference in writing tasks. However, these same qualities negatively correlate with user preference in STEM questions. Conversely, logical rigor is highly valued for STEM tasks but has little impact on preferences for writing tasks. Although we lack do not have dataset comparing individual user judgments, treating the users who ask STEM questions and those who ask writing questions as distinct groups provides some evidence of user-specific preferences.


| Vibe (low -> high)                                                                                                                                   | PP Coef     Writing    |        PP Coef     STEM          |
|:-----------------------------------------------------------------------------------------------------------------------------------------------------|:---------------:|:---------------:|
| Detail and Elaboration. Gives brief or shallow responses. -> Provides thorough, nuanced, and expansive information.                                   | **0.65**            | **-0.07**           |
| Humor and Playfulness. Responds in a straightforward and serious manner. -> Uses humor, playful language, or wordplay to make the response engaging. | **0.38**            | **-0.61**           |
| Formalness. Uses casual, conversational, or informal language. -> Uses formal and sophisticated vocabulary and sentence structure.                    | **-0.54**           | **0**               |
| Conciseness. Uses verbose language and excessive details. -> Uses minimal words to convey a point clearly.                                            | **0.32**            | **-0.06**           |
| Logical Rigor. Provides conclusions without thorough justification. -> Constructs well-supported arguments with clear reasoning.                     | **0.26**            | **0.82**            |
| Emotional Tone. Remains neutral or detached. -> Infuses responses with expressive emotion, making the tone enthusiastic or empathetic.               | **0.07**            | **-0.12**           | |

---

> ### Author Response · Authors · 2024-11-26
> **An overdue reply continued**
>
> ### Comparison with preset vibes
>
> While preset vibes achieve a similar preference prediction accuracy to the vibes identified by VibeCheck, this is not entirely negative. The primary goal of VibeCheck is not to build an interpretable reward model but to identify salient differences in a given pair of models that are informative of preference for a user or group of users. As shown in Section 5.2, VibeCheck uncovers more fine-grained vibes that better differentiate models compared to preset vibes. Moreover, these vibes are as influential as properties already known to significantly impact human preferences.
>
> While it is ideal for these vibes to also obtain a higher preference prediction accuracy, this is quite hard to for datasets like chatbot arena which have a diverse range of questions and users. Furthermore, while preset vibes are effective for preference prediction in settings with a diverse range of prompts, as demonstrated in Section 6, VibeCheck identifies vibes that are far more specialized to specific tasks. For example, when explaining the summarization differences between TNLGv2 and Command X, VibeCheck achieves superior model matching and preference prediction accuracy compared to preset vibes, as shown below:
>
> | Preset      |    Model Matching Acc    | Preference Prediction Acc |
> |-------------|:------------------------:|:--------------------------:|
> | Preset      |          69.07          |           58.08           |
> | VibeCheck   |          **71.29**          |           **61.42**           |
>
> Thanks to the reviewers we realize that we did not effectively communicate this in the main text and will update the manuscript accordingly.
>
> We have also updated the manuscript to include examples of the vibes obtained across experiments within the main text. We have also tried to better visualize the relationship between model matching and preference prediction with a nifty bar chart table. Please let us know if this is more clear than our previous version!

---

### Author Response · Authors · 2024-12-02
**Nearing the end of the discussion period**

Good afternoon!

As we are nearing the end of the discussion period, please let us know if there is anything else we can do to address your questions or concerns.

---

### Meta-Review · Area_Chair_BJr5 · 2024-12-17

**Metareview:**

The paper introduces VibeCheck, an innovative method for identifying and quantifying qualitative traits ("vibes") of LLM outputs, such as friendliness or formality, that are well-defined, differentiating, and user-aligned. By automating vibe discovery, validation, and alignment assessment, the authors address an important but underexplored aspect of LLM evaluation: subjective characteristics beyond correctness. Extensive experiments across tasks like summarization, math problem-solving, and image captioning demonstrate the method's ability to distinguish between popular LLMs (e.g., GPT-4, Claude, Llama3) and align discovered vibes with human preferences. While the study raises questions about the need for real-time vibe discovery versus predefined traits and lacks ecological validation, the paper provides a timely and user-centered framework for better understanding and evaluating LLM behaviors. I recommend acceptance with minor revisions to address these limitations (especially regarding the writing and task-dependent claim) and clarify the method's generalizability.

**Additional Comments On Reviewer Discussion:**

A major issue pointed out by several reviewers is the typos and clarity of the paper, which is nicely addressed through the rebuttal phase, and one reviewer raised the score. Another weakness a reviewer identified is the claim of user and task-dependent, which is also addressed during the discussion phase. Given the novel findings, this would be a nice paper if these changes were reflected in the final version.

---

### Decision · Program_Chairs · 2025-01-22

Accept (Poster)